# Transthoracic ultrasound localization microscopy of myocardial vasculature in patients

Jipeng Yan [1,6], Biao Huang [1,6], Johanna Tonko [2], Matthieu Toulemonde [1], Joseph Hansen-Shearer [1], Qingyuan Tan[1], Kai Riemer [1], Konstantinos Ntagiantas[3], Rasheda A. Chowdhury [3], Pier D. Lambiase[2], Roxy Senior [3,4,5] & Meng-Xing Tang [1] ✉

Myocardial microvasculature and haemodynamics are indicative of potential microvascular diseases for patients with symptoms of coronary heart disease in the absence of obstructive coronary arteries. However, imaging microvascular structure and flow within the myocardium is challenging owing to the small size of the vessels and the constant movement of the patient's heart. Here we show the feasibility of transthoracic ultrasound localization microscopy for imaging myocardial microvasculature and haemodynamics in explanted pig hearts and in patients in vivo. Through a customized data-acquisition and processing pipeline with a cardiac phased-array probe, we leveraged motion correction and tracking to reconstruct the dynamics of microcirculation. For four patients, two of whom had impaired myocardial function, we obtained super-resolution images of myocardial vascular structure and flow using data acquired within a breath hold. Myocardial ultrasound localization microscopy may facilitate the understanding of myocardial microcirculation and the management of patients with cardiac microvascular diseases.

Myocardial microvasculature is of substantial and increasing clinical importance. For example, for coronary heart disease (CHD), the leading cause of mortality worldwide, computer tomography coronary angiography (CTCA) is the recommended first-line investigation based on the most recent NICE guidelines[1], for patients with suspected angina or for those who are asymptomatic with suggested electrocardiogram (ECG) changes for ischaemia. However, CTCA assessment is limited to only large-artery obstruction, despite the fact that between 40–50% (refs. 2,3) of patients with symptoms of suspected CHD have ischaemia with non-obstructive coronary arteries (INOCA). This population includes a heterogenous group of patients with various aetiologies, including

coronary artery vasospasm and importantly, coronary microvascular disease (CMD), with the latter being evident in more than half of the patients with INOCA[4]. Compared with large-artery disease, our understanding of CMD is still fairly limited. Traditionally, CMD is diagnosed by the documentation of a diminished coronary flow reserve and the impaired ability of the microvasculature to augment blood flow in response to stress, which is indirectly evaluated from microvascular flow in arteries[5]. Initially, CMD was thought to be a combination of structural and functional changes at the level of the microvasculature; recent studies have led to a more differentiated understanding and subclassification to structural CMD and functional CMD subtypes[6,7]. They may both

[1]Ultrasound Lab for Imaging and Sensing, Department of Bioengineering, Imperial College London, London, UK. [2]Institute of Cardiovascular Science, University College London, London, UK. [3]Faculty of Medicine, National Heart and Lung Institute, Imperial College London, London, UK. [4]Royal Brompton Hospital, London, UK. [5]Northwick Park Hospital, Harrow, UK. [6]These authors contributed equally: Jipeng Yan, Biao Huang. ✉e-mail: mengxing.tang@imperial.ac.uk

manifest with stress perfusion defects on cardiac magnetic resonance imaging (MRI), yet differ on a pathophysiological level; hence, accurate characterization may have important clinical implications for patient management and for the choice of therapeutic interventions. CMD is also associated with non-ischaemic cardiomyopathy, inflammatory cardiac disease and even with non-obstructive coronary plaque disease[8].

Existing clinical modalities have challenges in imaging the microvascular structure and flow in the myocardium in vivo. Myocardial microvasculature refers to vessels within the heart wall, which are typically hundreds of microns or less in diameter[9]. Imaging is challenging owing not only to their small size but also their constant movement throughout the cardiac cycle. Both clinical CT, MRI and nuclear-imaging methods lack the spatial resolution and/or contrast to directly visualize these small vessels and the haemodynamics in them. Recently, ultrafast ultrasound[10,11] has shown promise in imaging such myocardial vessels ex vivo and in vivo. However, blood cells only generate weak scattering signals, limiting signal to noise ratios (SNRs), and image resolution is restricted by the classical wave-diffraction limit.

Ultrasound localization microscopy (ULM), a class of super-resolution ultrasound methods, can image deep microvasculature with resolution beyond the diffraction limit by accurately localizing microbubbles (MBs) from the contrast-enhanced ultrasound (CEUS) images[12–16]. Haemodynamics in the microvasculature can be measured in ULM by tracking the movements of super-localized MBs[17–20]. ULM has been used to image vasculature in many human body parts and organs, including the lower limbs[21], breast[22], liver[23] and brain[24]. The concept of using ULM for imaging coronary vasculature in myocardium has recently been demonstrated in small animals in two dimensions (2D) and 3D[25,26]. The 3D study showed the ability to assess myocardial infarction resulting from coronary artery disease. Microvascular disease is an increasingly recognized important differential diagnosis for ischaemia caused by epicardial coronary artery disease. Despite its clinical importance, assessments in humans have so far largely been restricted to indirect measurements of myocardial blood flow and coronary flow reserve. Direct visualization of myocardial vasculature at a microcirculation level, and haemodynamic quantification via ULM to investigate and characterize structural and functional microcirculation alterations would fill an important gap in contemporary cardiological diagnostics.

However, the clinical translation of ULM for non-invasive imaging of myocardial microvasculature remains a challenge, mainly owing to large cardiac motions and the required large field of view and penetration depth. ULM requires the detection and accumulation of MB localizations over time, assuming the vascular structure remaining stationary[15,16]. During the accumulation, any tissue motion, inevitable for in vivo imaging, would require correction to not degrade the image. Motion correction is a cross-cutting theme in medical-imaging technology, and in ULM some efforts have been made to correct for such motion (in particular for lower limbs[21], rat hearts[25,26] and rat kidneys[27]); however, in cardiac imaging, it is particularly challenging owing to large and complex cardiac motions. Furthermore, unlike in preclinical imaging, where a high-frequency probe (central frequency typically between 6–15 MHz) with an aperture size similar to that of the animal heart is used[25], clinical imaging would require a low-frequency probe (typically 1–4 MHz) to achieve appropriate penetration depth. In addition, because the human heart is larger and because of challenges in imaging through the ribs, the clinical aperture size will be much smaller than the heart. Such low frequency and small aperture, combined with diverging waves required for ultrafast acquisition, usually generate lower SNR and spatial resolution in deep tissue, posing additional challenges for the detection and tracking of MBs.

## Results

This work shows transthoracic myocardial ULM in human patients with reconstructed super-resolution vasculature and blood-flow dynamics in the myocardium, using a customized data-acquisition and processing pipeline (Fig. 1). In the pipeline, a pulse sequence is designed for low-frequency ultrasound (2 MHz) to detect and enhance CEUS signals with MB motion correction among steering angles; coherence to variance (CV) beamformer is incorporated to achieve high SNR and reduce side lobe artefacts; a two-level tissue motion correction strategy is proposed to deal with the inter- and intra-cardiac cycle motions; a feature-motion-model framework is developed to track MBs. The feasibility of the pipeline was demonstrated on two ex vivo Langendorff porcine hearts and on four patients (the first one with impaired left ventricular function from hypertrophic cardiomyopathy, the second with a dilated left ventricle secondary to high ventricular ectopy burden, the third with idiopathic ventricular fibrillation in a structurally normal heart and the fourth with hypertrophic cardiomyopathy with asymmetric septal hypertrophy). Parameters and components used for acquiring and processing in vivo data are briefly summarized in Supplementary Table 1.

A clinical cardiac phase array probe (M5Sc-D, GE HealthCare) was driven by a research ultrasound system (Vantage 256, Verasonics) at a central frequency of ~2 MHz. Diverging waves were transmitted at an ultrafast pulse repetition frequency of 5,490 Hz. Amplitude modulation (AM) was used for separating MB and tissue signals. By steering diverging waves in six angles, a frame rate of 305 Hz was achieved after compounding.

Ultrasound contrast agents used in this study included Sonovue (Bracco) MBs (concentration of ~$2 \times 10^8$ per millilitre), a sulfur hexafluoride gas core and a phospholipid molecule shell with size distribution indicated in Supplementary Fig. 1. We injected MBs in either infusion or a slow bolus, instead of a fast bolus, to avoid excessive MB concentrations in myocardium, as localizing and tracking MBs can be more challenging at higher concentration. For the ex vivo porcine heart imaging, MBs were infused by a syringe pump (Harvard Apparatus) at an infusion rate of 5 ml min$^{-1}$. For the in vivo human heart imaging, 2 ml of MBs (within the recommended clinical dose range) were manually injected in a slow bolus (~6 s). In this study, datasets were acquired on two ex vivo and four in vivo hearts. One ex vivo heart, denoted as Porcine 1, was scanned from both parasternal short- and long-axis views for 5 s each. The other ex vivo heart, denoted as Porcine 2, was scanned from parasternal short-axis view for 10 s. For in vivo hearts, denoted as Patients 1 to 4, we acquired 10 s data on parasternal short- or long-axis views, and the available views depended on the acoustic window of patients.

The processing diagram for reconstructing CEUS images is shown in Supplementary Fig. 2. To separate MB signals from tissue signals in the radiofrequency (RF) channel data, AM pulse sequence was first used, and the remaining tissue signals were further reduced by subtraction of the moving average across three frames. The B-mode and CEUS images were reconstructed in polar coordinates using the delay-and-sum (DAS) beamformer and the CV beamformer, respectively. A Doppler-based motion estimation[28] method was used to correct the movement of MBs among steering angles and enhance the contrast signals after angle compounding. Evidence of the improvement by the Doppler-based MB motion correction can be seen in Extended Data Fig. 1.

A two-stage—affine and B-spline-based non-rigid—image registration approach[21] was used to compensate for myocardium motions in the porcine heart. For the in vivo human dataset specifically, an image-intensity-based gating algorithm was first implemented to select and index frames in the diastolic phase, which had the least motions in the whole cardiac cycle, considering that the existing algorithm could not correct for out-of-plane motion via 2D ultrasound. The remaining motion of the myocardium within the diastolic phase in each cardiac cycle was corrected for using the two-stage image registration approach after removal of the MB signals from the tissue signals by singular value decomposition (SVD) filtering[29]. Finally, rigid image registration was used to align CEUS images across different cardiac

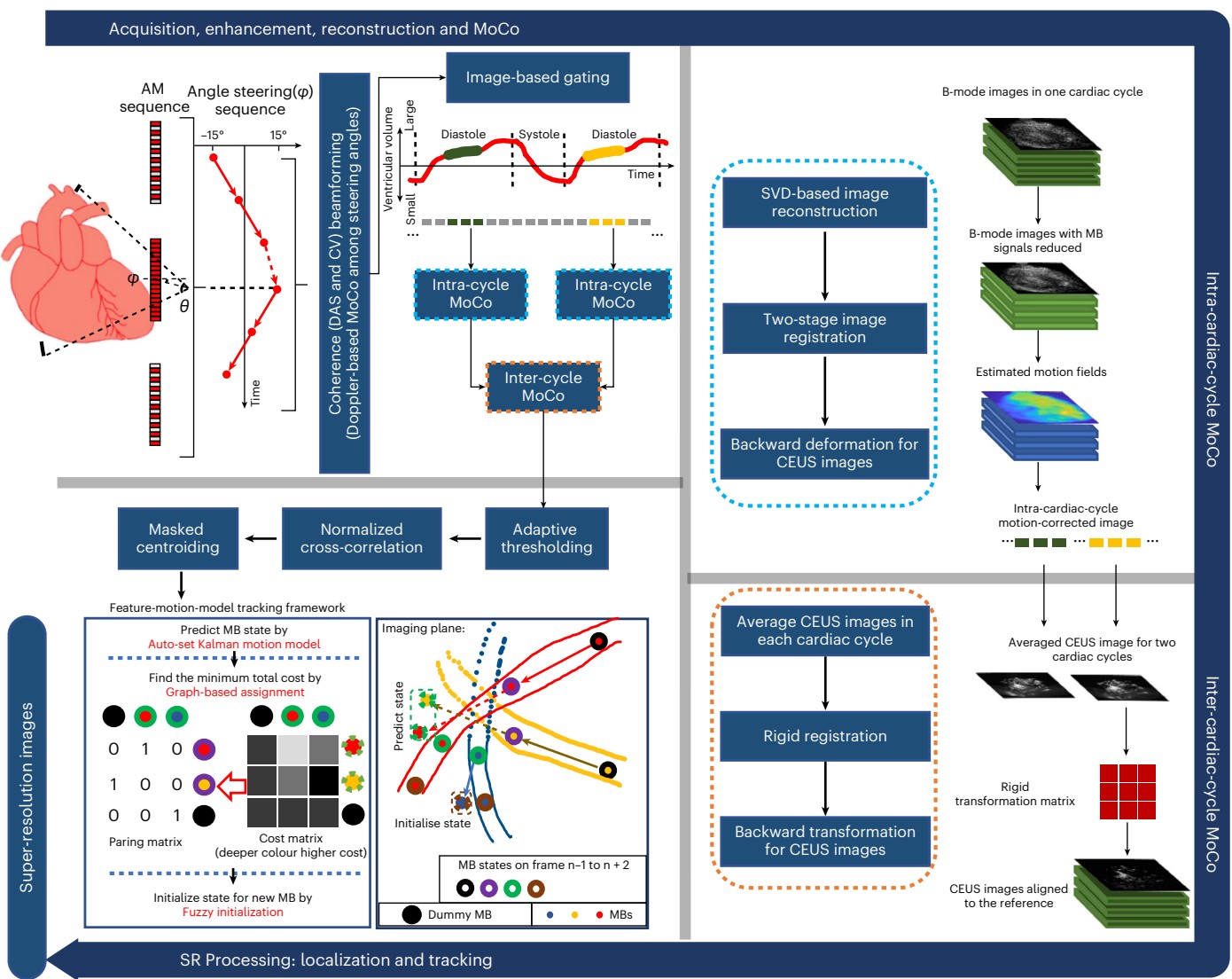

**Fig. 1 | In vivo data acquisition and processing pipeline.** A three-pulse AM sequence was used to detect and separate tissue and CEUS signals. Diverging waves were steered in six angles in a triangle sequence for angle compounding. MB movement between steering angles was detected and corrected to effectively enhance the contrast signal after angle compounding. Frames in diastole of the cardiac cycle were excluded by using a frame intensity correlation-based gating algorithm (top left). A two-level, intra-cardiac-cycle (top right) and inter-cardiac-cycle (bottom right), image registration strategy was proposed to correct for complex myocardium motions. MBs in the motion-corrected CEUS images were localized with normalized cross-correlation and paired by a proposed feature-motion-model tracking framework. Super-resolution (SR) images were plotted by accumulating MB trajectories (bottom left). Ex vivo data processing had the same pipeline without gating and inter-cardiac-cycle motion correction (MoCo).

cycles. The detected in-plane tissue deformation on the short-axis view between two frames during one cardiac cycle of a porcine heart and on the two views during the diastole phase of a human heart is similar and ranges between 1.4 mm and 1.7 mm. Comparisons of B-mode frames before and after motion correction are demonstrated in Supplementary Videos 1–3 for the three views. Effectiveness of this two-level motion correction strategy is also demonstrated when comparing Power Doppler images obtained with and without motion correction, as shown in Extended Data Fig. 2.

For each CEUS image, image patches containing bubble signals were segmented out by binary pixel connectivity after adaptive thresholding. Normalized cross-correlation was applied to each MB patch and the corresponding spatially varying point spread function (PSF) of the region where the MB signals are located. Single MB images were cropped out of the image patches with the mask on the basis of the cross-correlation coefficient map and the MB was localized through centroiding. The MBs were tracked to generate the flow dynamic information for the microvasculature and enhance the saturation of reconstructed ULM images with the MB trajectories. MBs were paired with our previously proposed feature-motion-model framework[30], where the ratio of the normalized image intensity difference between candidate MB pairs to the probability obtained from linear Kalman motion model with MB locations was set as a cost function, and MBs were paired by finding the total minimum with graph-based assignment[31]. The feature-motion-model framework was further improved in this study to track MBs. First, fuzzy initialization of the motion model was proposed to estimate the initial velocity of newly appeared MB to further improve the Kalman motion model performance. Second, a method to estimate the Kalman filter parameters from the data was also proposed to make the framework free from manual adjustment. Details of the tracking framework can be found in Supplementary Methods. MBs that were detected to appear in no more than 3 frames were regarded as low-confidence tracks and removed to improve tracking precision. Demonstrations of MB tracking can be found in

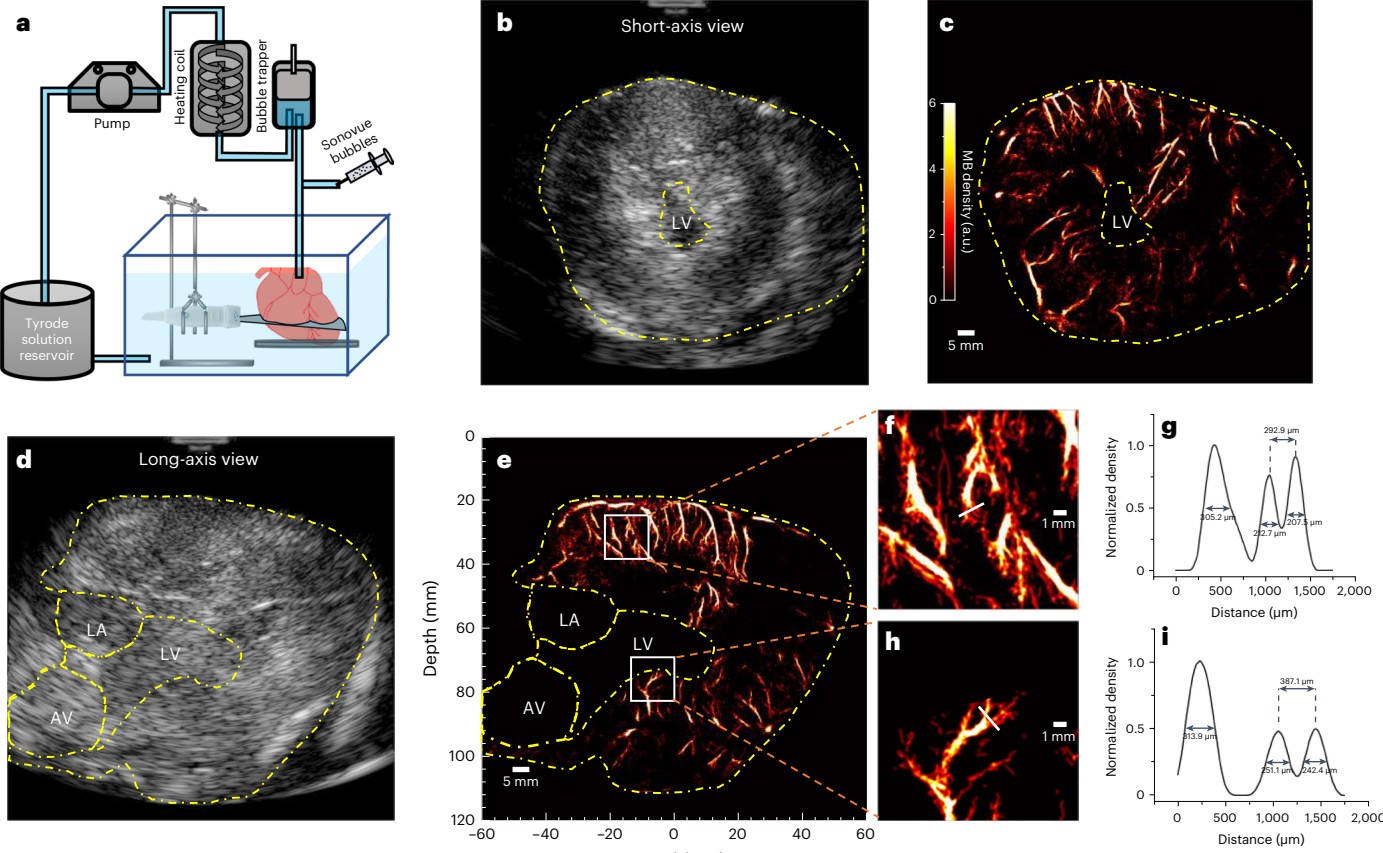

**Fig. 2 | Langendorff ex vivo heart-data acquisition and results for Porcine 1.**
**a**, The Langendorff setup for ex vivo dataset acquisition. **b**,**d**, The B-mode images from short- and long-axis views of the heart; the yellow dashed lines delineate the region cropped out in the processing. **c**,**e**, The corresponding SR density map for the microvasculature inside the myocardium. **f**,**h**, Zoomed-in density map of white boxes in **e**; white solid lines indicate where the vessel was cut for cross-section analysis. **g**,**i**, The density profile normalized to the maximum from the cross-section analysis.

Supplementary Videos 4 and 5. Super-resolution (SR) images were plotted by accumulating the MB trajectories in the map, each track having a width defined by a 2D Gaussian whose full width at half maximum (FWHM) was set as a quarter of a wavelength. Since the MB concentrations were too high for localization and tracking of individual bubbles inside the aorta, ventricle and atrium, these regions were cropped out in the analysis.

In the ex vivo porcine Langendorff heart experiment, the probe's position was adjusted and clamp fixed to have a good field of view for the short axis and long axis of the heart (Fig. 2a). From the B-mode images of the short- (Fig. 2b) and long-axis (Fig. 2d) views, myocardium surrounding the left ventricle (LV), left atrium (LA) and aortic valve (AV), delineated with dashed lines, can be visualized. The corresponding ULM MB density maps reconstructed from 5 s acquisitions are present alongside the B-mode images (Fig. 2c,e). Detailed microvasculature inside the left ventricle wall and the ventricular septum is presented in the density maps. MB density profiles along the white boxes in Fig. 2e magnified in Fig. 2f,h were measured and plotted after being normalized to the maximum (Fig. 2g,i). Vessels separated by distances of 292.9 μm and 387.1 μm can be distinguished from the profiles. The vessels' sizes were estimated using the FWHM and ranged from 207.5 μm to 313.9 μm. There is a dark region in the lateral area of Fig. 2e, while some vasculature below and above that area is still visible. Furthermore, the vasculature in the same position in the second ex vivo heart, as shown in Extended Data Fig. 3, is clearly visible. Therefore, it is likely that the apical vasculature in that area in ex vivo porcine heart 1 was damaged or blocked.

In the clinical acquisition, the probe was manually positioned by a clinician to acquire parasternal short- and long-axis views in the

patients. Positions of the imaging plane are shown in Fig. 3b. Both views were acquired for Patient 1, but only short-axis views were acquired for the other three patients due to the long-axis view being limited by the acoustic window of the three patients. After gating, a total of 1,110 frames (including 10 cardiac cycles), 990 frames (including 9 cardiac cycles), 630 frames (including 9 cardiac cycles), 930 frames (including 10 cardiac cycles) and 996 frames (including 12 cardiac cycles) within the diastolic phase were used to reconstruct ULM images in the short- and long-axis views of Patient 1 and the short-axis views of Patients 2, 3 and 4, respectively.

The microvasculature inside the interventricular septum and the left ventricular wall of the first patient is visible from the ULM MB density maps in both the long- and short-axis views (Fig. 3d,g). The closest equivalent views taken from multiplane reconstructions of the corresponding diastole CTCA scans are shown in Fig. 3a,c. Compared with the clinical diastole CTCA images, ULM exhibits increased sensitivity and resolution for myocardial microvessels. Some similar large vessels appear to be in both CTCA and ULM images (Fig. 3 white arrows). It should be noted that the CT images correspond to a slab with a constant thickness of 10 mm, while the ULM has a much smaller slab thickness (a typical cardiac probe has a slab thickness of one to a few wavelengths with varying depth, less than 5 mm for the used probe). Due to the slab thickness difference or the low SNR of ultrasound below the chamber, the vessel on the right side of the LV is presented in the CT image in Fig. 3a but missed in the ULM image in Fig. 3d. Myocardial vessels close to the top-right margin of heart in Fig. 3d might not be detected due to reduced probe sensitivity close to the left and right boundaries that affects both transmission and reception (shown in Supplementary

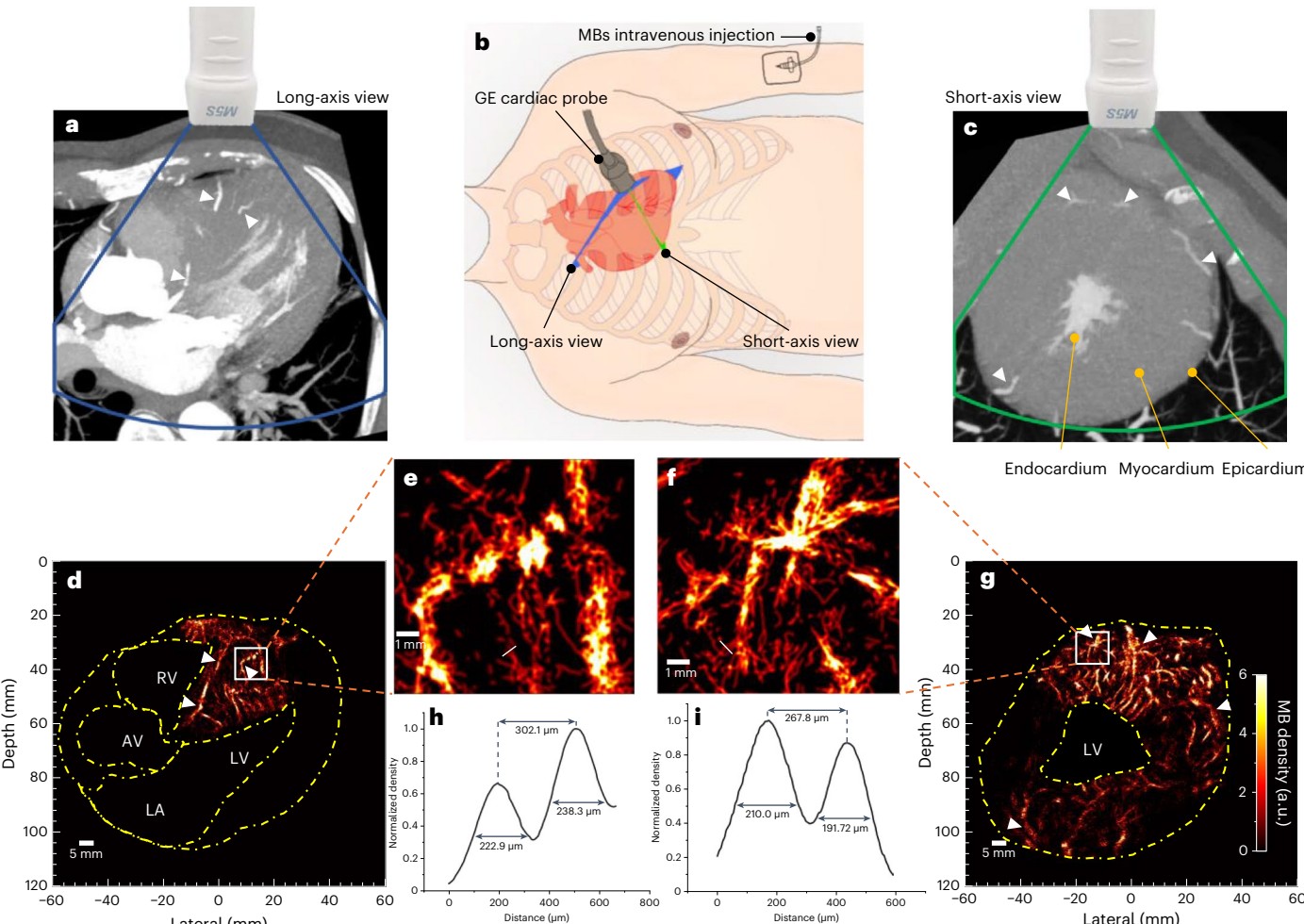

**Fig. 3 | In vivo CTCA scans and ULM imaging for Patient 1. a,c,** The long- and short-axis views in the CTCA scans approximated the equivalent plane as those in the ultrasound scans. White arrows indicate the similar structures that are visible in both modalities. **b,** Positions of the ultrasound probe on the chest for cardiac imaging. Blue and green planes indicate the long- and short-axis views of the heart, respectively. **d,g,** Myocardial ULM density maps. The yellow dashed lines indicate the chamber regions that were cropped out. **e,f,** Zoomed-in density map of white boxes in **d** and **g**, white solid lines indicate where the vessel was cut for cross-section analysis. **h,i,** The density profile normalized to its own maximum from the cross-section analysis. Overlapped images of the two modalities can be found in Supplementary Video 6.

Fig. 3). Many small vessels, from the epicardium penetrating the middle wall to the endocardium, are visible in the short-axis view (Fig. 3g), while in the CT scan only several major vessels were detected. Dilated myocardial vessels can be directly visualized from the reconstructed vasculature of Patient 1 diagnosed with hypertrophic cardiomyopathy. This type of cardiomyopathy is associated with a reduced coronary vasodilator reserve and increased resting coronary flow to meet the higher baseline oxygen demand due to the elevated myocardial mass and higher filling pressure[32]. Cross-section analysis was also performed on two visually separated vessels in the magnified regions (Fig. 3e,f). As shown in Fig. 3h,i, these two pairs of vessels are 302.1 μm and 267.8 μm apart, compared with half the wavelength of 320 μm. The MB density map for Patient 2 (Fig. 4b) also presents detailed microvasculature in the myocardium. Compared with the equivalent view by CT scan (Fig. 4a), two big vessels are also visible on the ULM results (white arrows in Fig. 4), but many vessels not visible in CT became clearly visible in the ULM results, as shown in the zoomed-in microvasculature density map (Fig. 4c,d). There was less reconstructed vasculature at the lateral sides than at the top and bottom parts in Fig. 4b. It may be because of the strong side lobes of the large and bright chamber. More vessels in the side wall are visible in Figs. 2c and 3g, and Extended Data Fig. 3 where the chambers are smaller. Reconstructed ULM images of Patients 3 and 4 can be found in Extended Data Fig. 4. In short, the visual

comparison demonstrates that ULM images present overall higher resolution and sensitivity to small myocardial vessels.

Image resolution was also estimated by a Fourier ring correlation (FRC)-based method through random splitting of MBs' whole tracks across frames in two subsets, following a previous study[33]. Based on the FRC estimation with ½ bit information threshold: the imaging resolutions for the ex vivo short- and long-axis views were 132 and 173 μm; the resolutions of the short- and long-axis views for Patient 1 were 149 and 202 μm; and the resolutions of the short-axis views for Patients 2 to 4 were 154 μm, 240 μm and 153 μm, respectively. FRC curves are provided in Fig. 5. The lateral resolutions of the contrast echocardiograph, as measured by the FWHM of the PSF, were 1,700 (1,950) and 5,850 (7,310) μm at depths of 30 and 120 mm for transmitted frequencies of 2.4 (1.7) MHz, respectively. The ULM image resolutions as estimated by FRC were just below half a wavelength of the transmission pulses (320 μm for 2.4 MHz and 452 μm for 1.7 MHz) and present at least 8-fold improvement with respect to contrast echocardiography. The variations in the estimated resolutions in different images using FRC are probably due to the low saturation level of the vascular images, hence causing uncertainties in FRC estimation[33].

Haemodynamics in the myocardium can be presented by flow-speed maps (Fig. 6a,d), which were obtained by tracking MB movements across frames in ULM processing, not available in conventional

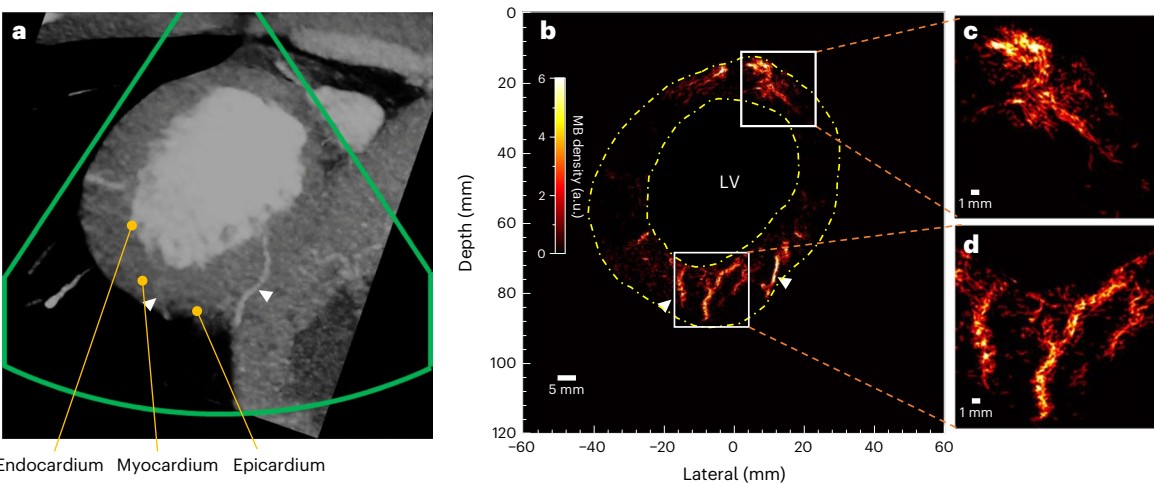

**Fig. 4 | In vivo CTCA scan and ULM images for Patient 2. a**, The short-axis view in the CTCA scan that corresponds to the ultrasound scan plane. White arrows indicate the similar structures that are visible in both modalities. **b**, The ULM density map. **c,d**, The zoomed-in ULM density maps for the boxed regions in **b**.

CTCA scans. Haemodynamics can also be presented in flow-direction maps (Fig. 6b,c), where the flow between epicardium and endocardium can be visualized. Animations of the reconstructed data with MB flowing inside microvasculature can be found in Supplementary Video 6. Two quantitative metrics, vessel diameter and flow-speed distributions (shown in Fig. 7), in the short-axis view of two patients were also obtained by the ULM technique.

## Discussion

We have shown the clinical feasibility of transthoracic ULM of myocardial microvascular structure and flow using a customized data acquisition and processing pipeline. Compared with CTCA, ULM shows more detailed myocardial microvasculature at diffraction resolution, can be accessed at bedside and is more affordable and free from ionizing radiation. The high resolution and sensitivity to myocardial vascular flow and its quantification potentially lead to better understanding of microcirculation in the myocardium.

The clinical ULM acquisitions were undertaken in a similar setup as that for echocardiograph examination in a cardiac outpatient clinic using an existing clinical echocardiography transducer, facilitating the clinical translation of the technique. The employed CE-marked contrast agent, Sonovue, was first approved in the European Union more than 20 years ago and is currently used in 36 countries. The intravenous bolus injection of the contrast agents is an established clinical operation for clinical contrast-imaging examination. The acquisition for ULM was done with a single ultrasound imaging system on a trolley, which is suitable for bedside investigation. One main limitation of the ULM technique is that currently, it is processed offline owing to the computational power demands of the hardware and the computational speed requirements of the algorithm, which can be optimized in the future.

Although ULM has revealed both macrovessels and microvessels in the myocardium in the four patients, these are only a small sample of the large vascular network in the myocardium, and there are regions with no apparent vessels identified. In this study, the total data-acquisition time was limited to ~10 s, which was dictated by the length of time that patients could hold their breath. Only data from the diastolic phase of the cardiac cycle with least myocardium motion was used for generating ULM images. Therefore, only vessels and flows perfused in this phase were reconstructed. There might be bias in measured flows. A frame rate of 305 Hz and a searching window size, set as 150 mm s$^{-1}$, used for tracking MBs would also mean that the fastest flow in large vessels could be missed. The 2D ultrasound can only detect the velocity projected to the imaging plane where the elevational component of velocity is lost. Temporal information was also lost in the ULM images owing to

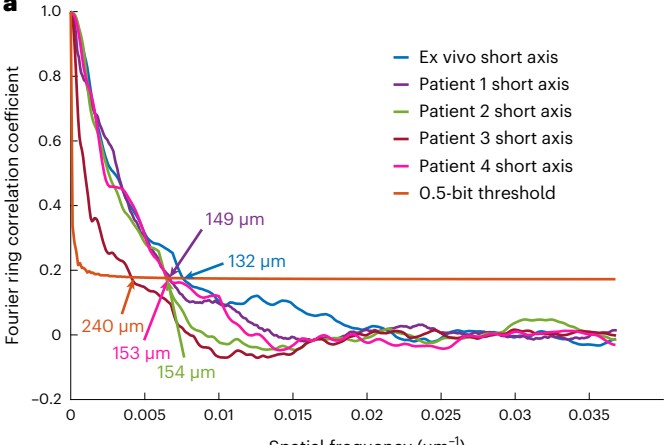

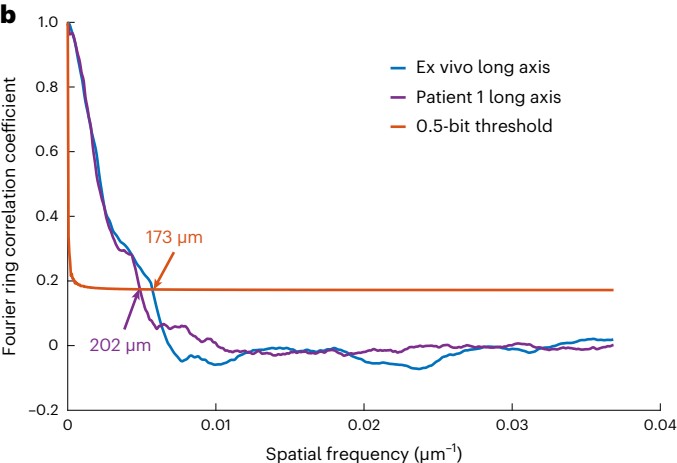

**Fig. 5 | FRC curves. a**, FRC evaluation on the short-axis view of the five datasets. **b**, FRC evaluation on the long-axis view of the two datasets. The arrows point to the crosses between the FRC curves and the half-bit threshold curves, and the numbers indicate the corresponding image resolution measured by the method.

the accumulation process in the reconstruction. To reveal more of the myocardial vasculature, there are important factors to consider. First, the data-acquisition time and number of CEUS image frames are very low in this case. The slice of the heart in the imaging plane can change

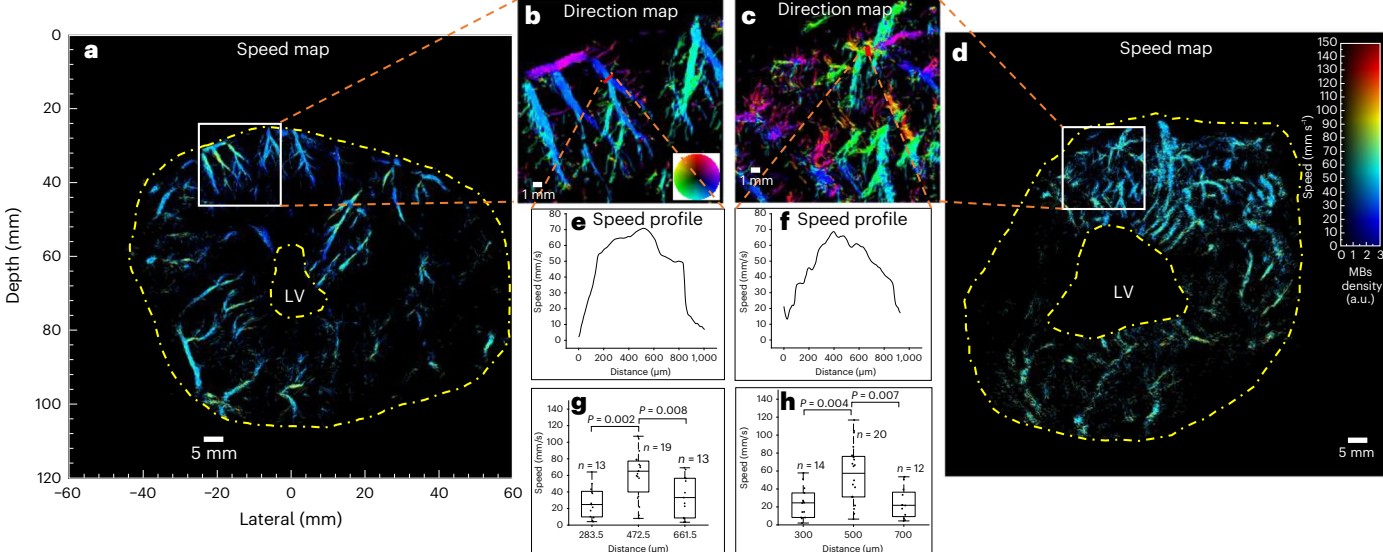

**Fig. 6 | ULM flow-speed and direction maps, and flow-speed profiles for an ex vivo porcine heart and a patient's heart, all in short-axis view. a,d,** Flow-speed maps corresponding to the SR density maps in Figs. 2 and 3. **b,c,** SR flow-direction maps of the boxed regions in **a** and **d**. **e,f,** Flow-speed profiles at the positions indicated by the red lines in **b** and **c**, generated by speeds on segments averaged across 0.5 mm width along the vessels. **g,h,** Flow-speed distribution of MBs passing through red lines in **b** and **c**. Measurements are for biological replicates. Bin width, 200 µm; black line, median; box limits, 25th and 75th percentiles; whiskers, minimum and maximum data points excluding outliers. The mean ± s.d. of the speed from left to right in **g**: 27.24 ± 18.71, 60.46 ± 28.00, 34.47 ± 24.30; **h**: 24.41 ± 16.98, 56.26 ± 32.32, 24.07 ± 16.85. Unpaired Mann–Whitney two-sided *U*-test was used. **e, f, g** and **h** demonstrate that large vessels have higher flow speeds at the centre than at the side.

drastically owing to breathing and to heart-twisting motions. Patients were asked to adjust their breath (to obtain appropriate imaging views of the heart according to clinician experience in conventional echocardiograph examination) and then hold the breath to reduce out-of-plane motion, which is important for 2D ultrasound acquisition and to restrict the acquisition time. Therefore, the number of frames available to us for cardiac ULM imaging was limited by the maximum time that patients could hold their breath and the available frames in the diastoles. In this case, the total acquisition time was up to 10 s, out of which only ~3 s of data during diastoles was used for reconstructing the final ULM image. The acquisition time of the cardiac data was orders of magnitude less than those typically used in brain ULM[14,24,34] where no breath holding and cardiac gating were required. This short acquisition time resulted in limited saturation of vasculature in the reconstructed cardiac ULM images, as shown in Supplementary Fig. 4, and made it challenging to distinguish impaired microcirculation from an incomplete mapping of vessels. The vessel-size measurements can be affected by the size of the 2D Gaussian used for plotting super-resolution images. When a small size is used, multiple tracks in a large vessel could be mistaken as multiple small vessels when the saturation is limited. A larger Gaussian used in plotting density maps can help reconstruct large vessels from separated trajectories, but will increase measured vessel size and can also merge separate small vessels that are close to each other. A trade-off is required, particularly when the vascular signal saturation is low. Three-dimensional imaging[25] with longer acquisition times and 3D motion correction could help address this issue (as discussed below).

Second, the sensitivity and specificity of detecting individual MBs with sufficient SNR are important and can be further improved. Compared with preclinical studies on small animals, the SNR of acquired clinical CEUS images is more challenging due to the increasing signal attenuation of the diverging waves over depth. Normalized cross-correlation smoothing of CEUS images with PSFs was chosen as the localization method to deal with limited SNR in CEUS images but reduced the probability of separating closely located bubbles. This could cause non-negligible errors in the reconstruction of microvessels with a diameter of a few hundred micrometres or those that are separated by a few hundred micrometres. Therefore, we used infusion and slow bolus to inject MBs in this study, rather than fast bolus, to reduce MB concentration and closely spaced MBs. Furthermore, there was still tissue breakthrough in the AM image that reduced the sensitivity of individual-MB detection. When moving-average background subtraction was used to remove the tissue breakthrough, signals from very-slow-moving MBs might be removed together with tissue signals. The sensitivity, specificity and SNR of the MB images could be improved by, for example, using nonlinear Doppler[35] when large tissue motion is present, as it is capable of separating tissue and bubble signals even when they are moving. Compared with other organs, the cardiac chambers filled with MBs further decrease the SNR and bubble specificity when ultrasound passes through, causing attenuation, phase aberration and nonlinear propagation. This may explain the darker ULM images in the lower part of the myocardium in Fig. 3g. Side lobes of chambers can also reduce contrast in MB images, as shown in Extended Data Figs. 2b and 5a, and thus reduce the sensitivity of localization near the chamber. Image reconstruction methods that can further reduce side lobes, especially those of chambers, are worth exploring. Optimization for improving MB detection can also be done during acquisition. For example, the transmitted pressure might need to be higher for patients with a higher body mass index (BMI), as their heart can be further away from the probe.

Future work to develop ultrafast 3D cardiac ultrasound can facilitate motion correction and further improve myocardial ULM. Correction for 3D cardiac motion is feasible if 3D + time imaging can be achieved using, for instance, a matrix-array transducer. With a whole cardiac cycle used in ULM, full-range flow speed can be reconstructed to show pulsatility[36], and temporal information of the whole cycle can be presented via a microbubble density cineloop[26], which might be useful for observation of the heart. This could enable much longer data acquisition while patients have shallow breathing. Furthermore, imaging frames within a longer duration of a cardiac cycle could be used if 3D motion is corrected for. Therefore, ULM images reconstructed with 3D imaging can be expected to achieve higher vascular saturation.

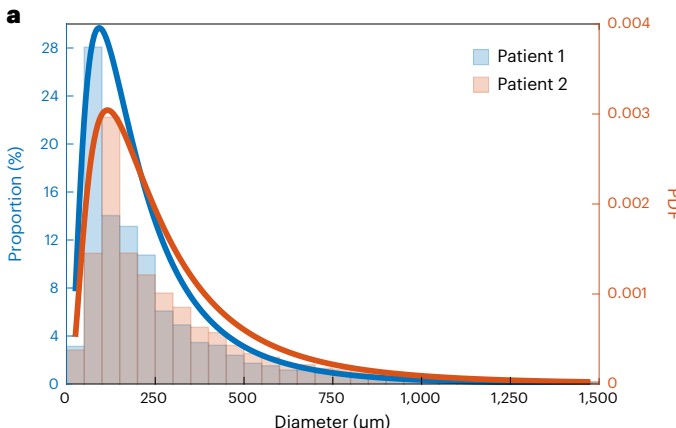

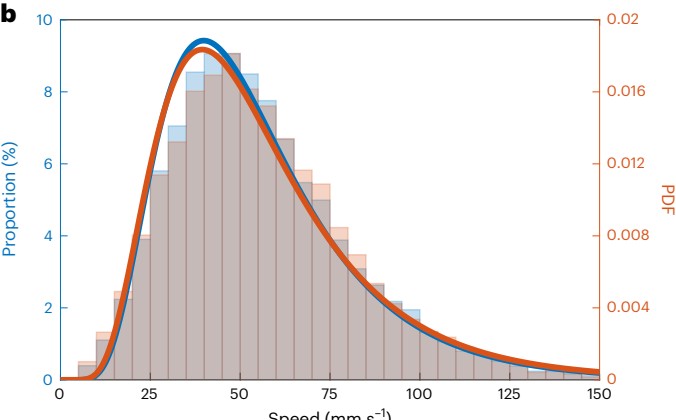

**Fig. 7 | Quantitative metrics obtained from SR images of the short-axis views of Patient 1 and Patient 2. a**, Proportion of vessel diameter distribution. Bin width, 50 μm. Vessel diameter larger than 1,500 μm were counted as 1,500 μm. Bin height was calculated as the ratio of the length of vessels with diameter in the corresponding bin range to the total vessel length. **b**, Proportion of speed distribution in the flow-speed maps of the short-axis view of the two patients. Bin width, 5 mm s⁻¹. Bin height was calculated as the ratio of the number of pixels with value in the corresponding bin range to the total number of pixels with non-zero values. Lines were created by fitting corresponding data with the log normal distribution, which has been used to describe the flow velocities in lymph node vessels[50]. PDF, probability density function.

Moreover, 3D imaging reduces the need to accurately position the transducer and improves the volumetric coverage of the imaged object. Covering a larger volume could make the quantification of myocardial vascular flow more robust.

## Methods

### Porcine heart and patient management
For the ex vivo experiment, two porcine hearts were explanted from two large white pigs (65–75 kg, 4–5 months old). Each heart was extracted as previously described[37] under anaesthesia using a human donor heart retrieval protocol. Animal studies were reviewed by the Royal Veterinary College Animal and Ethics Review Board and carried out in accordance with ethics standards (European Commission 2010, the Animal Welfare Act 2006 and the Welfare for Farm Animals (England) Regulations 2007).

Patient 1 was a 60-year-old male (BMI 24 kg m⁻², 167 cm, 67 kg) with hypertrophic cardiomyopathy and secondary prophylactic left-sided dual-chamber implantable cardioverter defibrillator (ICD). Patient 2 was a 31-year-old male (BMI 23.3 kg m⁻², 191 cm, 83.0 kg) with dilated left ventricular cavity (139.57 ml m⁻², but preserved left ventricular ejection fraction (LVEF) of 63%), no regional wall motion abnormalities, without evidence of myocardial scar in the cardiac MRI. The left ventricular

dilatation was probably secondary to a high burden of ventricular ectopy of >40%. Patient 3 was a 31-year-old male (BMI 25.3 kg m⁻², 175 cm, 77.6 kg) with a background of idiopathic ventricular fibrillation with structurally normal heart in cardiac MRI and secondary prevention transvenous dual-chamber ICD insertion, and who presented with appropriate ICD shocks due to recurrent ventricular ectopic triggered ventricular fibrillation. Patient 4 was a 34-year-old male (BMI 28.9 kg m⁻², 178.5 cm, 72.8 kg) with hypertrophic cardiomyopathy with asymmetric septal hypertrophy (maximum wall thickness 21 mm) without outflow tract obstruction, LVEF of 75% and primary prevention subcutaneous ICD. Cardiac MRI reported patchy late gadolinium enhancement throughout the hypertrophied septum as well as large dense perfusion defects throughout the septum in a perfusion study with adenosine.

All patients were recruited from the cardiac outpatient clinic after giving informed and written consent to participate in a prospective clinical cohort study investigating novel methods for non-invasive arrhythmogenic substrate characterization including high frame rate contrast-enhanced transthoracic echocardiography. The study was reviewed and approved by the London–Bromley Research Ethics Committee and the Health Research Authority (IRAS Project ID 144257, REC reference 14/LO/0360) and is ongoing.

### Acquisition
**Coronary CT angiography.** The prospective ECG-gated CT coronary angiography scans were performed using a Siemens SOMATOM scanner. Following administration of sublingual glyceryl trinitrate (GTN) and intravenous metoprolol, an axial cross-section of the heart was acquired with patients in the supine position upon inspiratory breath hold.

**Ultrasound sequence.** The acquisition of ultrasound images for ex vivo and in vivo heart was performed using a Vantage Verasonics acquisition system and a phased-array ultrasonic probe (GEM5ScD, GE; central frequency, 2.84 MHz; 90% bandwidth at −6 dB; pitch, 0.27 mm; and 80 ×3 elements, one row in centre and two 80 half-height elements at sides). A transmit frequency of 2.4 MHz was used for the ex vivo experiment and Patient 1, and 1.7 MHz was used for Patients 2 to 4. The ultrafast ultrasound virtual source of the diverging wave was set at 21.6 mm above the centre of the probe, giving a 53° angular field of view. The imaging sequences consisted of the emission of 6 angles of diverging waves, in ascending (−15°, −3°, 9°) and descending (15°, 3°, −9°) triangle sequence[28], which was designed for Doppler-based motion estimation. Sonovue MBs were used as the ultrasound contrast agents. The gas-encapsulated MBs compress and expand asymmetrically when exposed to ultrasound waves. Such nonlinear echo-response can be exploited for separating MB signals from tissue signals with the AM sequence in each steering angle. While the Verasonics system took time to change its output voltage and there was potential nonlinearity in the output voltage of the system and in the response of piezoelectric elements, the ultrasound fields with full or half amplitudes in the AM sequence were not generated by exciting the probe with different voltages. Instead, a three-pulse AM sequence with half, full and half amplitude was used. The ultrasound fields with half amplitude were generated by dividing the transducer elements into two interleaving subgroups, and exciting each half seperately as shown in Fig. 1, to achieve improved linear cancellation and reduce the impact of cross-talk between neighbouring elements[38]. The pulse central frequency and transmitted pressure were between 50 and 90 kPa at the depth of between 30 and 50 mm, as demonstrated in Supplementary Fig. 5. The transmitted pressure was chosen to obtain good SNR without destroying many bubbles. The pulse length was kept as short as possible to maintain spatial resolution. Two different transmit frequencies, 1.7 and 2 MHz, were used. The lower frequency has higher bubble detection sensitivity, but 2 MHz has slightly better resolution. The probe was

fired at the maximum pulse repetition frequency (5,490 Hz) allowed by the desired imaging depth of 120 mm. With angle compounding and coded AM transmission, the frame rate was $5,490 \div 6 \div 3 = 305$ Hz.

Acquisitions (5 s and 10 s) were made on the ex vivo heart and on patients, respectively. For each acquisition, backscattered echoes were received by the phase array probe and digitalized by the Verasonics system at four times the transmitted centre frequency. RF data for each channel were stored for offline image reconstruction and processing in MATLAB (R2021a, MathWorks) on a desktop (CPU, AMD Ryzen 9 5900 Processor; GPU, Nvidia GeForce RTX3080; RAM, 128 GB).

### Image reconstruction

The echoes of the three pulses in the AM sequence were combined in each channel to generate the contrast-specific AM signals. Probably due to nonlinear propagation[39], tissue signals could not be completely cancelled out by the AM sequence. To reduce the residual tissue signals, a temporal moving average subtraction was applied to RF channel data across three frames for each steering angle on the basis of the assumption that tissue remains static but MBs move within the temporal averaging window, which is ~10 ms.

Hilbert transform was used to obtain analytic channel signals. B-mode images were reconstructed using the delay-and-sum beamforming method with coherent angle compounding[40], keeping the linearity of image intensity to meet the requirement of the SVD that was used in tissue motion correction. CEUS images were reconstructed using the CV beamforming method[41] to reduce the side lobes and noise, benefitting the MB isolation for super-localization. The CV beamformer calculates an adaptive weight for each pixel using the ratio of the squared coherent sum to the variance across all the channels and all the steering angles. The pixel value was then generated by the multiplication of the adaptive weight and the coherent sum.

Multi-angle compounding assumes stationary MBs between the steering angles, while in reality MBs move and hence the compounded MB signals decorrelate[42]. In this study, a Doppler-based method[28] was used to estimate MB velocities among the steering angles and the CV beamformer was then implemented after correcting for MB signal shifts. Note that the phase rotator used in literature[28] was discarded when correcting for the signal shifts, as Hilbert transform, instead of in-phase and quadrature demodulation, was used here. Images were reconstructed in polar coordinates (grid: 67.8 μm and 0.5°) for MB motion correction, envelope detected and transferred in Cartesian coordinates (grid: 67.8 μm along depth and 135.0 μm laterally) with spline interpolation for tissue motion correction and tracking in ULM processing.

### Image-based tissue motion correction

In vivo tissue motions were corrected for at two levels, intra-cardiac-cycle and inter-cardiac-cycle, considering the complexity of heart beats and the limitation of 2D ultrasound. Since out-of-plane motion correction was challenging for 2D ultrasound due to loss of signals, we focused on the diastole, where twisting motion is the least among all the phases in a cardiac cycle.

We applied an image intensity-based gating algorithm to the dataset. The cardiac cycle presents a periodic change in pixel intensity distribution between frames, which can be tracked from the change in the normalized cross-correlation coefficient between two adjacent frames. The frame with the lowest correlation coefficient appearing during one cardiac cycle indicates the start of systole. By automatically finding the different phases of a cardiac cycle, we were able to only take data during the diastoles (that is, between 0.2 and 0.4 s of data) for further analysis and ULM image generation.

To correct for intra-cardiac-cycle tissue motions, a two-stage—affine and then B-spline-based non-rigid[43]—image registration was conducted on the B-mode images to estimate and correct for tissue motions[21]. A reference image for motion correction was obtained by finding the frame with maximal similarity, measured using the MATLAB 'ssim' function, to the average frame of a cycle. B-mode ultrasound images in each cardiac cycle were processed using the 'svd' MATLAB function and reconstructed with the 5% largest singular values to reduce the effect of moving MB signals on motion estimation. Reconstructed B-mode images were log compressed with a dynamic range of 50 dB and rescaled to grey scale. The sum of square pixel intensity difference was used as the image difference for both the affine and non-rigid image registration. The grid of B-spline control points was set with a spacing of 16 (for ex vivo) or 32 (for in vivo) pixels along both the lateral and depth dimensions. Unrealistic non-rigid deformation might be generated after registration when speckle patterns changed with out-of-plane deformation and wave interference. The thin-plate spline constraint was used as a penalty to restrict the distortion[43]. Affine and non-rigid registrations were solved using the Levenberg–Marquardt and the steepest decent algorithms, respectively. The maximum iteration steps for affine and non-rigid transformation were both set at 500. Each stage of the optimization would stop earlier if the image difference between two adjacent iterations changed less than 0.001% for 20 times. The first-stage optimization would also stop earlier when the difference between transformation parameters in two iterations was less than $1 \times 10^{-4}$, for example, ~$4.4 \times 10^{-6}$ wavelength for translation along depth. The two-stage image registration began from the reference frame, and transformation matrices estimated on the current frame were set as the starting point for the next frame, reducing the computation time and improving temporal smoothness of the motion fields.

Following the correction for intra-cardiac motion, the inter-cardiac-cycle motions were subsequently corrected for through a rigid transformation using CEUS images that had clear anatomical signals from chambers and large vessels useful for registration. First, CEUS images were averaged in each cardiac cycle to reduce the image difference due to flowing MBs, and rigid transformation was then estimated between the averaged CEUS images to correct for motions between cardiac cycles. The averaged CEUS images were also log compressed and rescaled to grey scale before registration. Note that while out-of-plane motion cannot be ruled out, they would bring additional information from neighbouring slices of the myocardium.

For the ex vivo dataset, the tissue motions were corrected for by applying the two-stage image registration to all the acquired frames, as there was no breathing, probe motion or heavy twisting motion of the heart during the acquisition.

### Super-resolution processing

Adaptive thresholds were obtained using the 'adaptthresh' MATLAB function for each frame. Pixels with intensity lower than the adaptive thresholds or the estimated noise level were set to zero to reduce noise and side lobes in the CEUS images. MBs patched in each frame were generated using the connectivity of non-zero pixels. Considering the spatially varying PSFs in images generated using diverging wave, the image was divided into 5 (depth) × 5 (lateral) regions and PSF in each region was estimated by averaging 10 single MB images that were segmented from the CEUS sequence. PSFs estimated from ex vivo data are presented in Extended Data Fig. 6. Normalized cross-correlation[19] was calculated between each patch of MBs and the PSF in the region where the patch centre was. Single MB images were cropped out of the patches by the mask of the cross-correlation coefficient map whose value was over 0.5. MBs were localized by the pixel-intensity-weighted centroid of each MB image and rounded to a pixel map with a resolution of 13.5 μm, and the intensity of an MB was given by the sum of pixel values in the corresponding single MB image. The localization uncertainties of the used method were measured to be no more than 16 μm by imaging a wire target at three different depths in water, as demonstrated in Supplementary Table 2. MBs were tracked in a feature-motion-model framework[30]. MBs in two frames were paired by finding the global minimum of costs in the graph-based assignment[31,44]. The cost of a candidate MB pair was defined as the ratio of the image intensity difference normalized by the

stronger intensity of the two paired MBs to the probability obtained by a linear Kalman motion model[17,45–49] established using the positions of MBs. An MB could be paired to a dummy MB if its pairing costs to any other MBs were over a limit or there were different numbers of MBs in the two frames. The MB paired to the dummy MB was regarded as either a disappeared or newly appeared MB. In addition, the fuzzy initialization method was proposed to initialize the Kalman state vector of a newly appeared MB with estimated velocity, instead of assuming the MB to be static initially, to further improve the performance of the motion model. A parameter estimation method was proposed to automatically estimate the covariance of model prediction noise to avoid human interaction in setting up the motion model and to avoid the effort of tuning the parameter. Explanations of the MB tracking problem with Shannon theory and detailed descriptions of the tracking framework can be found in Supplementary Methods. A comparison on in vivo data between the proposed method and the 'SimpleTracker' used in previous studies[24,34] can be found in Extended Data Fig. 7.

### Display (SR maps and animation)
MBs tracked over at least four frames were kept for plotting images. Trajectories of flowing bubbles were plotted by linking paired MB positions using straight lines. The localization density map was plotted by accumulating trajectories with a width defined by a 2D Gaussian, using the 'imgaussfilt' MATLAB function, with FWHM set as a quarter of the ultrasound wavelength at the transmitted centre frequency to enhance image saturation due to the limited acquisition time. One density map plotted directly using the pixel size is presented in Extended Data Fig. 8.

To plot the flow-speed map, MB moving speeds were accumulated on each pixel of the trajectory. Accumulated speed and accumulated localization density maps were both filtered by a disk whose diameter is the same as the Gaussian FWHM. The flow-speed map was calculated by dividing the disk-smoothed accumulated speed by the disk-smoothed accumulated density, instead of directly applying smoothing on an unsmoothed speed map, to reduce the effect of smoothing kernel size on velocity measurement. To draw the flow-direction map, accumulated MB velocities were smoothed by the same disk and then divided by the disk-smoothed density map. The flow direction on each pixel was calculated from the averaged velocity direction. Flow-dynamics maps are presented by hue-saturation-value (HSV) colours with hue to illustrate speed or direction and value to illustrate MB density.

Animated flow was rendered by moving each localized bubble position along its trajectory with the estimated speed. Warm or cold colour indicates MB moving upwards or downwards.

### Parameter calculation
Vessel diameters were measured using vessel centre lines and boundaries without segmenting vessels. The centre lines of vessels were detected on the binarized density map using the 'bwmorph' function in MATLAB with 'skel' property set to 'inf'. The 'bwdist' function was applied to the inversed binarized density map to obtain the distance between the centre line and the nearest boundary of the vessel. The vessel diameter was defined by doubling the distance corresponding to each pixel on the centre lines. The vessel distribution corresponding to a certain diameter was counted using the sum of pixels on centre lines for all vessels of this diameter. Distributions of blood flow speeds were calculated from the non-zero-pixel values on the SR speed maps.

### Reporting summary
Further information on research design is available in the Nature Portfolio Reporting Summary linked to this article.

### Data availability
A sample-image dataset and step-by-step instructions for using the data are available via Figshare at https://doi.org/10.6084/m9.figshare.22331635 (ref. 51). Raw data samples are available from the SRUS Software Github repository at https://github.com/JipengYan1995/SRUSSoftware. Source data are provided with this paper.

### Code availability
SRUS executable software containing data processing and graphical user interface is available on GitHub at https://github.com/Jipeng-Yan1995/SRUSSoftware. Estimated PSFs, localization uncertainty and maximum blood-flow speed for searching MBs used for processing the data used in this study were set as default parameters in version 2.1. A sample-image dataset for testing localization and tracking is provided via Figshare at https://doi.org/10.6084/m9.figshare.22331635 (ref. 51).

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

## Acknowledgements

We thank the Royal Veterinary College and J. Perkins for assistance with the porcine-heart extraction; D. Panagopoulos, D. Agha-Jaffar, S. Li and C. R. Gonzalez for help with the ex vivo heart experiment; L. Gvinianidze and R. Hampson for suggestions to the in vivo acquisitions; M. Lerendegui for helping with code development; D. Garcia for the explanation of the Doppler-based motion-correction technique. M.-X.T. discloses support for the research described in this study from the Chan Zuckerberg Foundation (grant number 2020-225443), the Engineering and Physical Sciences Research Council (grant number EP/T008970/1 and EP/V04799X/1) and the National Institute for Health Research i4i (grant number NIHR200972). R.A.C. discloses support for the research described in this study from Rosetrees Trust (grant number M645 to R.A.C.). B.H. and Q.T. disclose support for studentship from the China Scholarship Council and the Imperial College Department of Bioengineering PhD Scholarship. K.N. discloses support for studentship from the Wellcome Trust (grant number 222845/Z/21/Z). P.D.L. is supported by UCL/UCLH Biomedicine NHR.

## Author contributions

M.-X.T. conceived the study. J.Y. and M.-X.T. designed the data processing pipeline. R.A.C. and K.N. designed the Langendorff ex vivo setup. M.-X.T., B.H., M.T., J.T. and J.Y. designed the in vivo setup. M.T. measured and calibrated the cardiac probe. M.T. and B.H. developed the acquisition sequence. B.H., K.N., M.T., K.R. and Q.T. acquired ex vivo data. J.T., B.H. and J.H.-S. acquired in vivo data. M.T. and J.Y. developed the image reconstruction algorithm. B.H. and J.Y. developed the motion correction algorithm. J.Y. and M.-X.T. developed the super-resolution imaging framework and software. B.H. and J.Y. processed the data. B.H., J.T., J.Y., M.-X.T., P.D.L. and R.S. interpreted the results. J.Y., B.H. and M.-X.T. wrote the first draft. All authors edited and approved the final version of the manuscript.

## Competing interests

M.-X.T. is a co-founder and shareholder of CardioAcc company, and a shareholder of Sonalis Imaging company. R.S. receives speaker fees from Bracco (Milan, Italy), Lantheus Medical Imaging (Boston, USA), GE Healthcare St Giles (Amersham, UK) and Philips Healthcare Ltd (Eindhoven, the Netherlands). The other authors declare no competing interests.

## Additional information

**Extended data** is available for this paper at https://doi.org/10.1038/s41551-024-01206-6.

**Correspondence and requests for materials** should be addressed to Meng-Xing Tang.

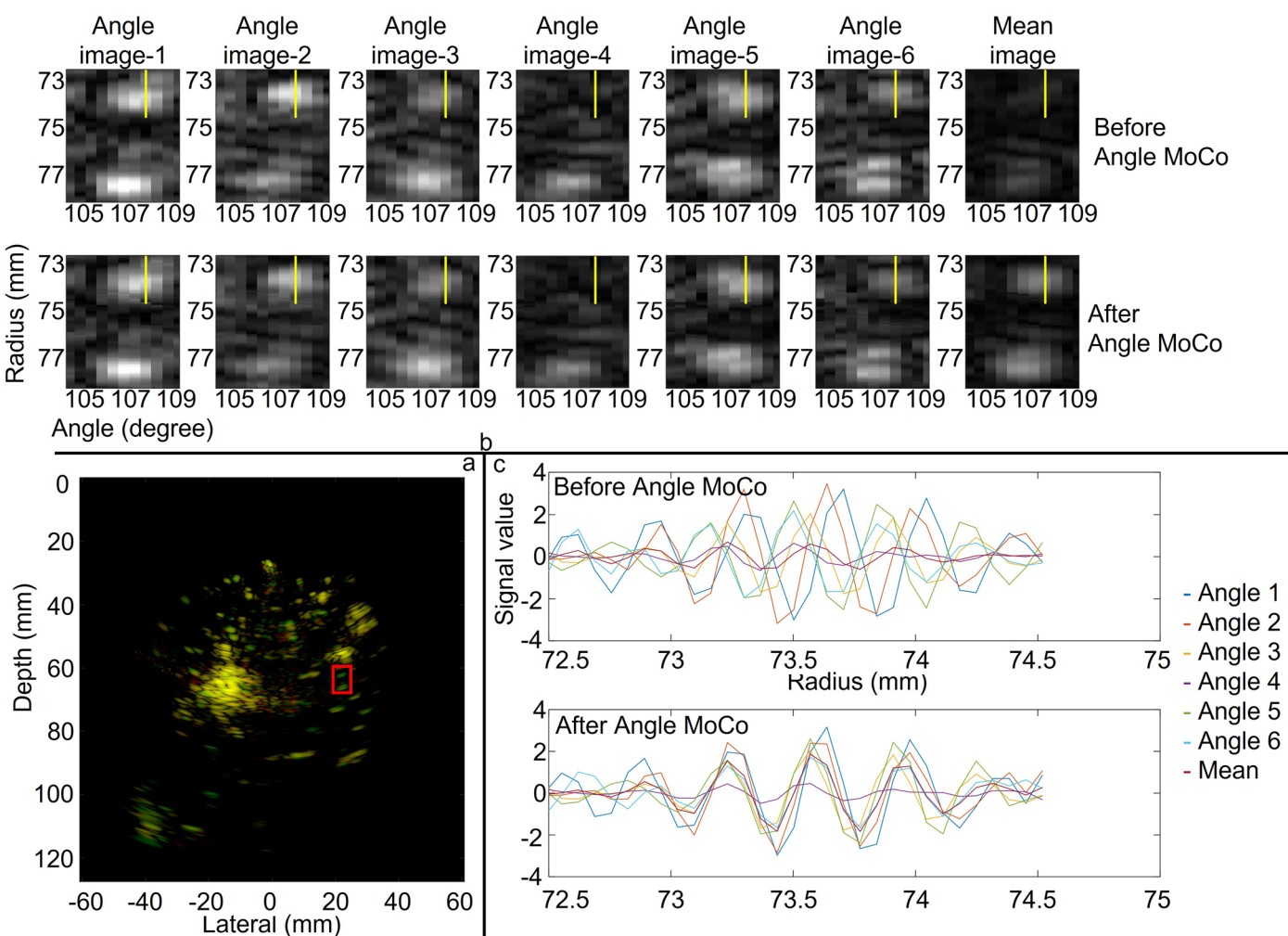

**Extended Data Fig. 1 | Improvement in bubble detection using Doppler-based motion correction before multi-angle compounding, noted as Angle Moco for convenience. a**, overlay of compounded images reconstructed by delay and sum (DAS) with and without Angle Motion Correction (MoCo) were normalized, log compressed with dynamic range of 40dB, and then the overlay was done by Matlab 'imfuse' function, using red for the image without Angle MoCo, green for the image with Angle Moco, and yellow for areas of similar intensity between the two images. Green colour covers more area, particularly in areas of isolated bubbles. **b**, magnified image of the region in the box in **a**, and plotted in polar coordinates, linear scale, and same dynamic range, after envelope detection. Compounded image obtained by averaging angle image with Angle MoCo presents stronger intensity than its counterpart. **c**, the real part of the signals along the yellow line in **b** before and after MoCo. Signal shifts among steering angles were corrected to achieve a higher coherence for compounding.

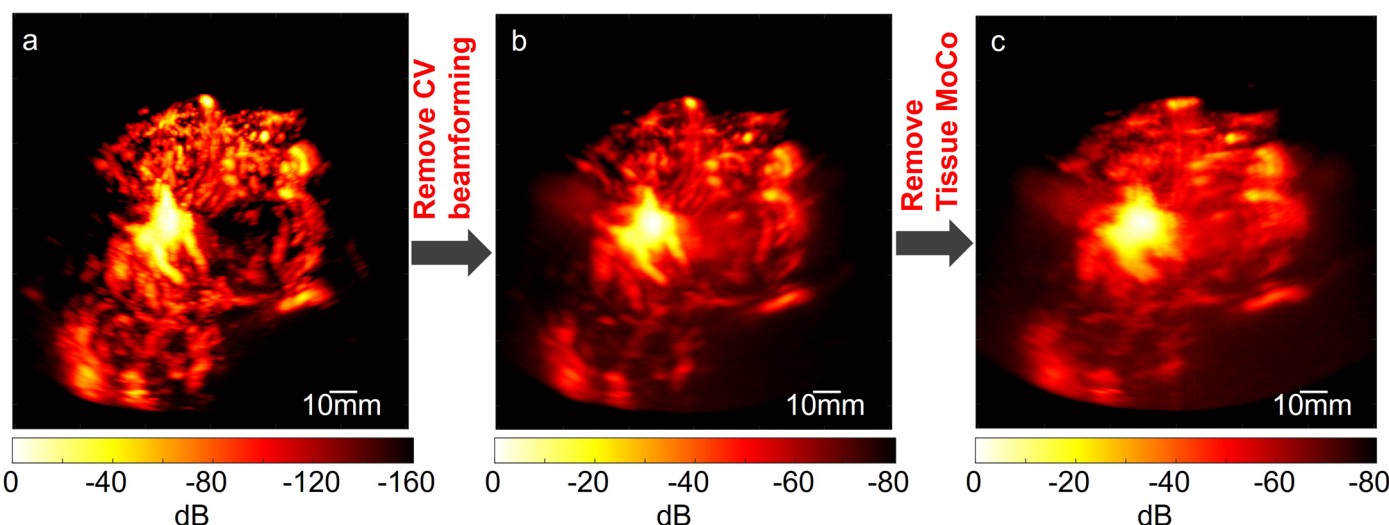

**Extended Data Fig. 2 | Power Doppler images of CEUS sequence acquired from the short-axis view of Patient 1 and reconstructed with different methods.** **a**, Power Doppler (PD) image of the CEUS sequence used for SR processing. **b**, PD image of the CEUS sequence obtained by replacing our CV beamforming with the conventional DAS beamforming and coherent compounding. By comparing **a** and **b**, CV beamforming can significantly reduce side lobes and noise. **c**, PD image of the CEUS sequence by removing Tissue Motion Correction (MoCo) from **b**. The vessels were significantly blurred without Tissue MoCo.

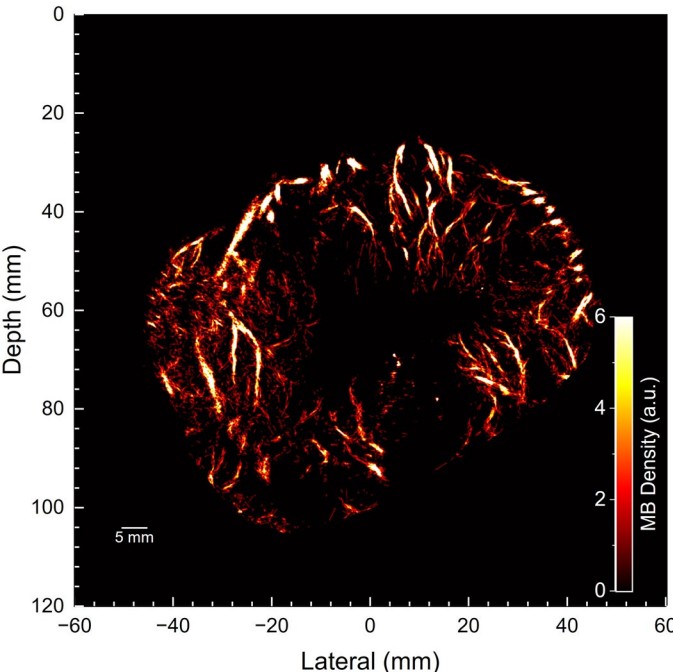

**Extended Data Fig. 3 | ULM image reconstructed from data acquired from Langendorff ex vivo porcine heart 2.** The data was acquired from the parasternal short-axis view for 10s.

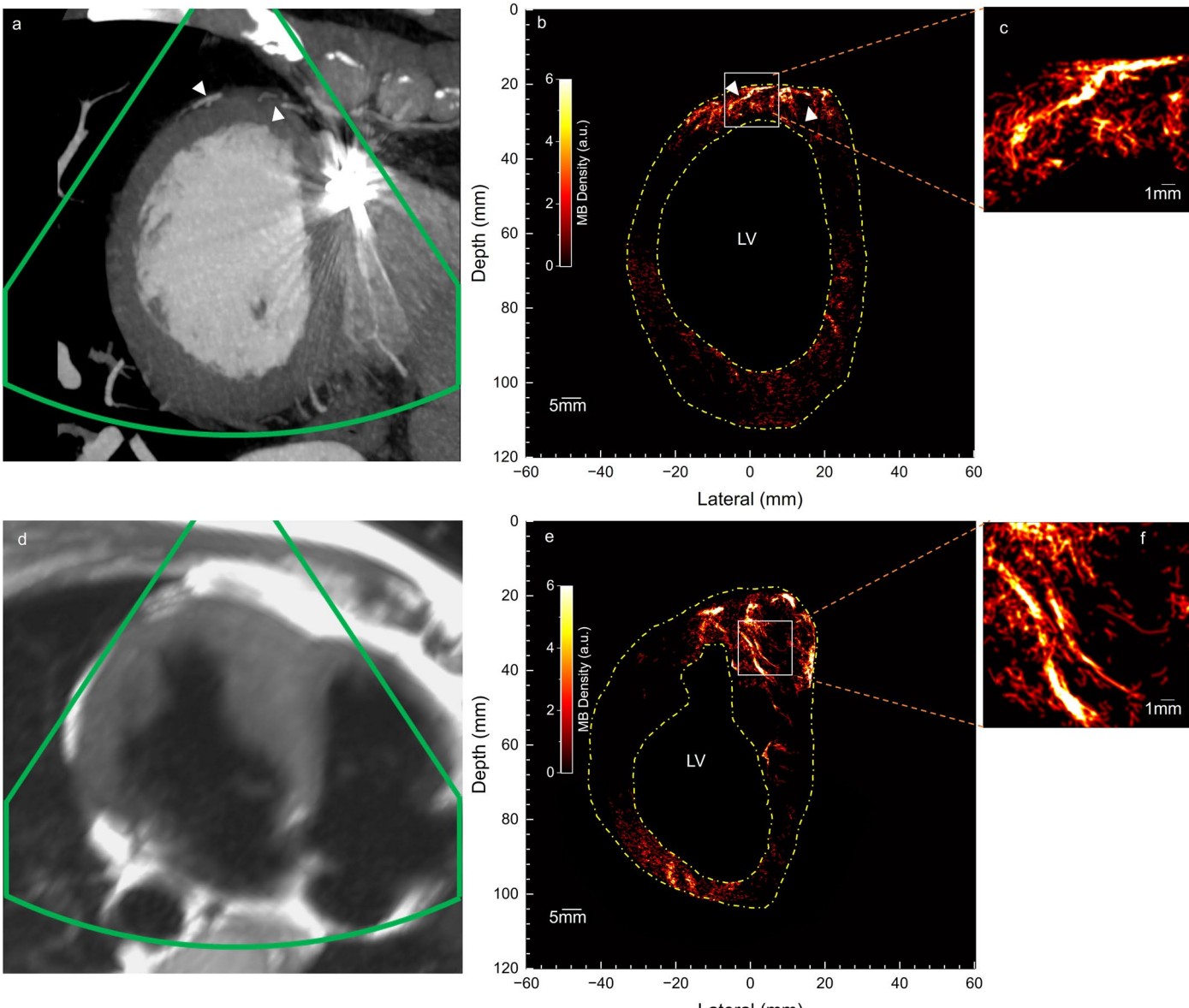

**Extended Data Fig. 4 | *In vivo* clinical CT / MRI scans and ULM images for Patient 3 and 4. a**, The short-axis view in the CTCA scan that corresponds to the ultrasound imaging plane. White arrows point out similar structures visible in both modalities. **b**, The ULM density map of the myocardial microvasculature. **c**, The zoomed-in ULM density map for the box region in **b. d**, The short-axis view in the Cardiac MRI scan that corresponds to the ultrasound imaging plane. **e**, The ULM density map of the myocardial microvasculature. **f**, The zoomed-in ULM density map for the box region in **e**. Note that the bright reflection in the CTCA of Patient 3 was generated from the lead of the implantable cardioverter defibrillator, which interfered with CTCA imaging. Patient 4 did not have clinical indication for a CTCA scan, so only MRI scan was conducted. The Cardiac MRI scan with perfusion was not acquired at the same slice as the ultrasound. One slice of 3D Cardiac MRI sequence, suppressing signal from flowing blood and maintaining high signal in the surrounding stationary tissues, is used here to illustrate the myocardium.

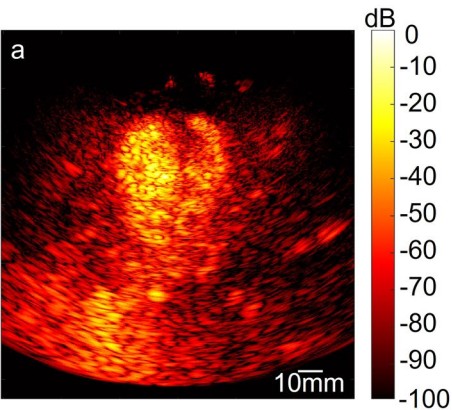 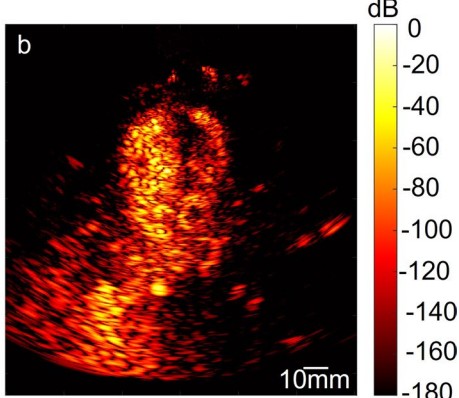

**Extended Data Fig. 5 | Power Doppler images obtained by beamforming data acquired from Patient 2 with DAS and CV beamformers. a**, Power Doppler image reconstructed by DAS beamformer. **b**, Power Doppler image reconstructed by CV beamformer. Strong contrast signals from within the ventricle chamber can generate significant side lobe artefacts on the side wall, which can be seen from Extended Data Fig. 3b,5a. Side lobes of the chamber can significantly affect the detection of MBs. The CV beamformer can help reduce side lobes but may also reduce the intensity of microbubble signals at the side lobe area. This may be the reason why less vessels are seen on the side wall of patient two.

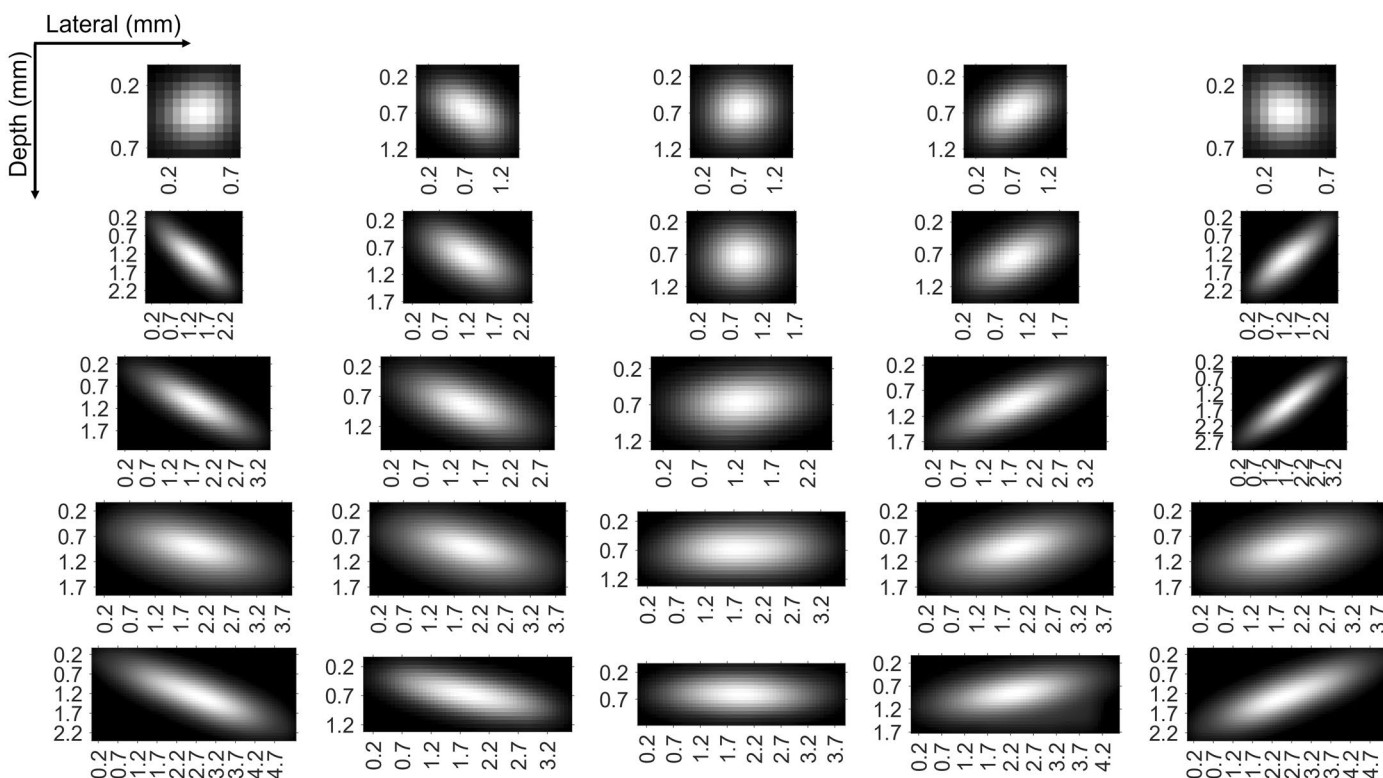

**Extended Data Fig. 6 | PSFs estimated from the processed ex vivo CEUS data.** Note that the PSF was estimated after thresholding out low intensity pixels, resulting in smaller estimated PSFs at image lateral sides where signal intensity was low due to the element directivity. The patches were cropped to the estimated PSF tightly to save computation in normalised cross-correlation.

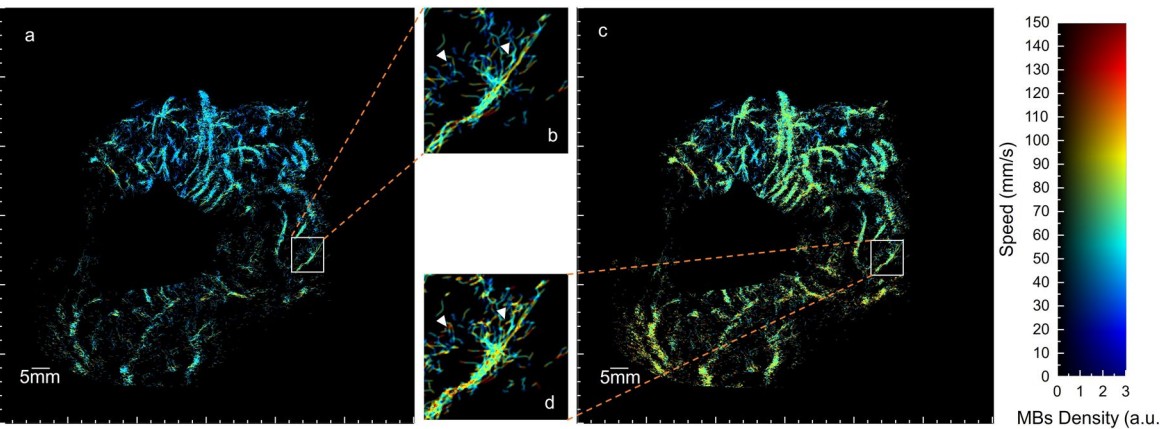

**Extended Data Fig. 7 | Velocity magnitude map obtained by our proposed method and the 'SimpleTracker'.** 'SimpleTracker(tinevez/simpletracker MathWorks)' was set with no gap filling, the same searching window size, 150 mm/s, and the same filtering, only accepting MBs with persistence of more than 3 frames. **a**, map obtained by our tracking framework. **b**, magnified image of the region in the box of **a**. **c**, map obtained by the 'SimpleTracker'. **d**, magnified image of the region in the box of **c**. **b** provides visually clearer vasculature than

**d**. 'SimpleTracker' pairs MB by finding total minimum of cost that is defined with distance between MBs. It prefers to pair closer MBs but also accepts pairs with larger distance only if two MBs are within the searching window and satisfy the topology constraint. The trajectories, pointed by white arrows in **d** may be erroneous as they are between vessel branches and with speed much higher than the surroundings. These trajectories were rejected by the proposed method, as motion model worked as an additional constraint.

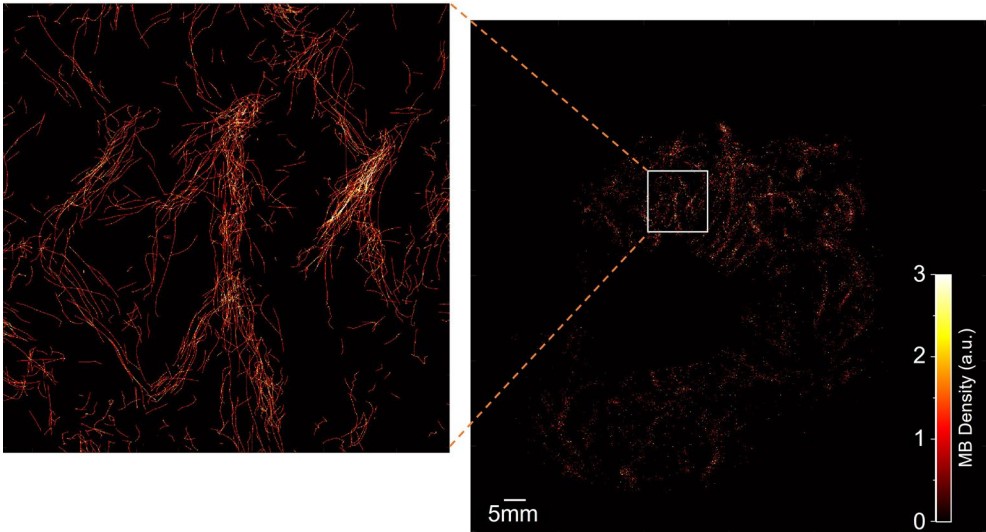

**Extended Data Fig. 8 | ULM density map of the short-axis view of Patient 1 without smoothing and the magnified image of the box region.** The pixel size used for reconstructing the ULM density map is 13.5 um, which is approximately 1/60 of wavelength. By painting only one pixel per localization in the ULM density map, vessels were far from being saturated due to the small pixel size and the limited acquisition time. The selection of a quarter of the wavelength for generating ULM images was used in the rest of the study to improve the visibility of the microvessels in the ULM image.

# Reporting Summary

## Statistics

For all statistical analyses, confirm that the following items are present in the figure legend, table legend, main text, or Methods section.

| n/a | Confirmed | |
|---|---|---|
| ☐ | ☒ | The exact sample size (*n*) for each experimental group/condition, given as a discrete number and unit of measurement |
| ☐ | ☒ | A statement on whether measurements were taken from distinct samples or whether the same sample was measured repeatedly |
| ☐ | ☒ | The statistical test(s) used AND whether they are one- or two-sided<br>*Only common tests should be described solely by name; describe more complex techniques in the Methods section.* |
| ☒ | ☐ | A description of all covariates tested |
| ☒ | ☐ | A description of any assumptions or corrections, such as tests of normality and adjustment for multiple comparisons |
| ☐ | ☒ | A full description of the statistical parameters including central tendency (e.g. means) or other basic estimates (e.g. regression coefficient) AND variation (e.g. standard deviation) or associated estimates of uncertainty (e.g. confidence intervals) |
| ☐ | ☒ | For null hypothesis testing, the test statistic (e.g. *F*, *t*, *r*) with confidence intervals, effect sizes, degrees of freedom and *P* value noted<br>*Give P values as exact values whenever suitable.* |
| ☒ | ☐ | For Bayesian analysis, information on the choice of priors and Markov chain Monte Carlo settings |
| ☒ | ☐ | For hierarchical and complex designs, identification of the appropriate level for tests and full reporting of outcomes |
| ☒ | ☐ | Estimates of effect sizes (e.g. Cohen's *d*, Pearson's *r*), indicating how they were calculated |

*Our web collection on statistics for biologists contains articles on many of the points above.*

## Software and code

Policy information about availability of computer code

| | |
|---|---|
| Data collection | Custom data-acquisition codes developed in Matlab2022a are available on reasonable request. |
| Data analysis | Code for the Doppler-based motion correction method was proposed in J. Porée, D. Posada, A. Hodzic, F. Tournoux, G. Cloutier and D. Garcia, "High-Frame-Rate Echocardiography Using Coherent Compounding With Doppler-Based Motion-Compensation," in IEEE Transactions on Medical Imaging, vol. 35, no. 7, pp. 1647-1657, July 2016.<br><br>We used existing B-spline free-form registration. A CPU code version can be found in Matlab Central File Exchange (https://www.mathworks.com/matlabcentral/fileexchange/20057-b-spline-grid-image-and-point-based-registration). We developed a CUDA version ( V11.4.120) for faster computation.<br><br>Executable super-resolution software for general super-resolution processing containing all key SRUS processing steps, as well as a graphical user interface, can be found on GitHub at https://github.com/JipengYan1995/SRUSSoftware. The software consists of data loading, motion correction, super-localization, tracking, plotting, parameter calculation, and animation generation. Detailed descriptions and user guides of the software can be found in the repository. |

For manuscripts utilizing custom algorithms or software that are central to the research but not yet described in published literature, software must be made available to editors and reviewers. We strongly encourage code deposition in a community repository (e.g. GitHub). See the Nature Portfolio guidelines for submitting code & software for further information.

## Data

Policy information about availability of data

All manuscripts must include a data availability statement. This statement should provide the following information, where applicable:

- Accession codes, unique identifiers, or web links for publicly available datasets
- A description of any restrictions on data availability
- For clinical datasets or third party data, please ensure that the statement adheres to our policy

A sample-image dataset and step-and-step instructions for using the data are available from figshare at https://doi.org/10.6084/m9.figshare.22331635. Raw data samples are available from the SRUSSoftware Github repository at https://github.com/JipengYan1995/SRUSSoftware.

## Research involving human participants, their data, or biological material

Policy information about studies with human participants or human data. See also policy information about sex, gender (identity/presentation), and sexual orientation and race, ethnicity and racism.

| Reporting on sex and gender | Sex and gender were not considered in the study design. |
|---|---|
| Reporting on race, ethnicity, or other socially relevant groupings | Socially relevant variables were not considered in the study design. |
| Population characteristics | Four patients were recruited from the cardiac outpatient clinic after giving informed and written consent to participate in a prospective clinical cohort study investigating novel methods for non-invasive arrhythmogenic substrate characterization including high-frame-rate contrast-enhanced transthoracic echocardiography.<br><br>Patient 1 was a 60-year-old male (BMI 24 kg/m2 – 167 cm, 67 kg) with typical hypertrophic cardiomyopathy and with a secondary prophylactic left-sided dual chamber implantable cardioverter defibrillator (ICD). (Fig. 3, Fig. 5b).<br><br>Patient 2 was a 31-year-old male (BMI 23.3 kg/m2 – 191 cm, 83.0 kg) with dilated left ventricular cavity (139.57ml/m2, but preserved left ventricular ejection fraction (LVEF): 63%), no regional wall motion abnormalities, without evidence of myocardial scar in the cardiac MRI. The left ventricular dilatation was likely secondary to high burden of ventricular ectopy of > 40%. (Fig. 4).<br><br>Patient 3 was a 31-year-old male (BMI 25.3 kg/m2 – 175 cm, 77.6 kg) with background of idiopathic ventricular fibrillation with structurally normal heart in cardiac MRI and secondary prevention transvenous dual chamber ICD insertion who presented with appropriate ICD shocks due to recurrent ventricular ectopic triggered ventricular fibrillation. (Extended Data Fig.4a,b,c).<br><br>Patient 4 was a 34-year-old  male (BMI 28.9 kg/m2 - 178.5cm, 72.8kg) with hypertrophic cardiomyopathy with asymmetric septal hypertrophy (max. wall thickness 21mm) without outflow tract obstruction, LVEF 75%, and a with a primary prevention subcutaneous ICD. Cardiac MRI reported patchy LGE throughout the hypertrophied septum as well as large dense perfusion defects throughout the septum during a perfusion study with adenosine. (Extended Data Fig.4e,f,g) |
| Recruitment | Patients were recruited from the cardiac outpatient clinic after giving informed and written consent to participate in a prospective clinical cohort study investigating novel methods for non-invasive arrhythmogenic substrate characterization including high-frame-rate contrast-enhanced transthoracic echocardiography. Criteria to select the patients in this study include a good acoustic window and signal-to-noise ratio in the CEUS data judged visually by the operator. Such selection shouldn't impact the conclusions of the study, as this is a feasibility study (with no statistics comparing patient groups, for example). |
| Ethics oversight | The study was reviewed and approved by the London-Bromley Research Ethics Committee and the Health Research Authority (IRAS Project ID 144257, REC reference 14/LO/0360) and is ongoing. |

Note that full information on the approval of the study protocol must also be provided in the manuscript.

# Field-specific reporting

Please select the one below that is the best fit for your research. If you are not sure, read the appropriate sections before making your selection.

☒ Life sciences ☐ Behavioural & social sciences ☐ Ecological, evolutionary & environmental sciences

For a reference copy of the document with all sections, see nature.com/documents/nr-reporting-summary-flat.pdf

# Life sciences study design

All studies must disclose on these points even when the disclosure is negative.

| | |
|---|---|
| Sample size | No sample-size calculation was performed as this study does not present any statistical analysis between patients. The study shows the feasibility of transthoracic super-resolution ultrasound localization microscopy in all the patients (N=4) with comparison to available CTCA or cardiac MRI. A sample size of 4 is appropriate for proof of concept. |
| Data exclusions | No data were excluded. |
| Replication | For the proof-of-concept experiments, myocardial vasculature was successfully reconstructed from four patients. The ultrasound probe was held by an experienced clinician to target the imaging plane. The starting time of acquisition needed to be delayed after microbubble injection to await for the microbubbles to perfuse the myocaridum. Codes and software were validated by simulations, phantoms and ex vivo experiments before clinical experiments. |
| Randomization | Randomization was not relevant to the study, as it only aimed to show the feasibility of transthoracic super-resolution ultrasound localization microscopy in patients. All the patients were allocated in a single group, and no statistical analyses between patients were carried out. |
| Blinding | Blinding was not relevant to the study. |

# Reporting for specific materials, systems and methods

We require information from authors about some types of materials, experimental systems and methods used in many studies. Here, indicate whether each material, system or method listed is relevant to your study. If you are not sure if a list item applies to your research, read the appropriate section before selecting a response.

## Materials & experimental systems

| n/a | Involved in the study |
|---|---|
| ☒ | ☐ Antibodies |
| ☒ | ☐ Eukaryotic cell lines |
| ☒ | ☐ Palaeontology and archaeology |
| ☐ | ☒ Animals and other organisms |
| ☒ | ☐ Clinical data |
| ☒ | ☐ Dual use research of concern |
| ☒ | ☐ Plants |

## Methods

| n/a | Involved in the study |
|---|---|
| ☒ | ☐ ChIP-seq |
| ☒ | ☐ Flow cytometry |
| ☒ | ☐ MRI-based neuroimaging |

## Animals and other research organisms

Policy information about [studies involving animals](); [ARRIVE guidelines]() recommended for reporting animal research, and [Sex and Gender in Research]()

| | |
|---|---|
| Laboratory animals | Two porcine hearts were explanted from large white pigs (65–75kg, 4–5-months old). |
| Wild animals | The study did not involve wild animals. |
| Reporting on sex | Sex and gender were not considered in the study design. |
| Field-collected samples | The study did not involve samples collected from the field. |
| Ethics oversight | The animal studies were reviewed by the Royal Veterinary College Animal and Ethical Review Board, and carried out in accordance with ethical standards (European Commission 2010, the Animal Welfare Act 2006 and the Welfare for Farm Animals (England) Regulations 2007). |

Note that full information on the approval of the study protocol must also be provided in the manuscript.

