## [Peer Review File · Nature Biomedical Engineering]

Transthoracic ultrasound localization microscopy of myocardial vasculature in patients

Corresponding author: Meng-Xing Tang

Editorial note

This document includes relevant written communications between the manuscript's corresponding author and the editor and reviewers of the manuscript during peer review. It includes decision letters relaying any editorial points and peer-review reports, and the authors' replies to these (under 'Rebuttal' headings). The editorial decisions are signed by the manuscript's handling editor, yet the editorial team and ultimately the journal's Chief Editor share responsibility for all decisions.

Any relevant documents attached to the decision letters are referred to as **Appendix #**, and can be found appended to this document. Any information deemed confidential has been redacted or removed. Earlier versions of the manuscript are not published, yet the originally submitted version may be available as a preprint. Because of editorial edits and changes during peer review, the published title of the paper and the title mentioned in below correspondence may differ.

Correspondence

Mon 15 May 2023

Decision on Article nBME-23-0750

Dear Prof Tang,

Thank you again for submitting to *Nature Biomedical Engineering* your manuscript, "Transthoracic super-resolution ultrasound localization microscopy of myocardial vasculature in patients". The manuscript has been seen by 3 experts, whose reports you will find at the end of this message.

You will see that the reviewers appreciate the work. However, they express concerns about the degree of support for the claims, and provide useful suggestions for improvement. We hope that with significant further work you can address the criticisms and convince the reviewers of the merits of the study. In particular, we would expect that a revised version of the manuscript provides:

- * An in-depth analysis of resolution and flow velocity, as suggested by reviewer #2.
- * Assessment of bubble-localization uncertainties and how these might cause errors in the reconstruction of the microvessels, as requested by reviewer #1.
- * Discussion of possible incomplete mapping of vessels due to short acquisition times, and how this may affect the results obtained, as suggested by reviewer #2.
- * Discussion of the upper and lower flow-velocity limits that can be detected with the current pipeline, and the impact of vessel orientation on the results, as suggested by reviewer #2.
- * Ideally, imaging evidence in additional porcine hearts, as suggested by reviewer #1.
- * Discussion, supported by evidence, of the limitation of the techniques, and whether the findings were in agreement with the clinical diagnosis, as suggested by reviewer #3.* Thorough methodological details, as per the relevant comments of all reviewers.

When you are ready to resubmit your manuscript, please upload the revised files, a point-by-point rebuttal to the comments from all reviewers, the reporting summary, and a cover letter that explains the main improvements included in the revision and responds to any points highlighted in this decision.

Please follow the following recommendations:

* Clearly highlight any amendments to the text and figures to help the reviewers and editors find and understand the changes (yet keep in mind that excessive marking can hinder readability).

* If you and your co-authors disagree with a criticism, provide the arguments to the reviewer (optionally, indicate the relevant points in the cover letter).

* If a criticism or suggestion is not addressed, please indicate so in the rebuttal to the reviewer comments and explain the reason(s).

* Consider including responses to any criticisms raised by more than one reviewer at the beginning of the rebuttal, in a section addressed to all reviewers.

* The rebuttal should include the reviewer comments in point-by-point format (please note that we provide all reviewers will the reports as they appear at the end of this message).

* Provide the rebuttal to the reviewer comments and the cover letter as separate files.

We hope that you will be able to resubmit the manuscript within 20 weeks from the receipt of this message. If this is the case, you will be protected against potential scooping. Otherwise, we will be happy to consider a revised manuscript as long as the significance of the work is not compromised by work published elsewhere or accepted for publication at *Nature Biomedical Engineering*.

We hope that you will find the referee reports helpful when revising the work. Please do not hesitate to contact me should you have any questions.

Best wishes,

Liqian

Dr Liqian Wang
Associate Editor, Nature Biomedical Engineering

Reviewer #1 (Report for the authors (Required)):

In this work, the authors have demonstrated the feasibility of using transthoracic ULM for investigating the myocardial microvascular hemodynamics on one large animal model and two patients, but only during the diastolic phases.

The topic of imaging myocardial microvessels is interesting and it definitely opens up more feasibilities for a broader range of clinical applications of ULM. Addressing the motion correction issues of imaging the heart sections with dramatic 3D motion is challenging without volumetric imaging, yet the authors have set a starting point by using available image registration techniques to correct motions during diastolic phases. This manuscript demonstrates the use of super-resolution ultrasound images for cardiovascular vascular imaging. One ex vivo experiment was performed with a porcine heart and two patient was imaged and the authors has demonstrated the intra-cardiac-cycle and inter-cardiac-cycle tissue motion correction to overcome the Influence of heartbeat. However, the overall image quality was poor comparing with other

SRUS image. There are some concerns as follow:

1. Instead of tracking the bubbles, which takes a long time for data processing, what does the power Doppler with contrast agents look like for the myocardial microvessels?
2. In lines 304-305, why haven't the authors used EKG signals for identifying the cardiac cycles?
3. In lines 309-310, this two-stage image registration method seems very similar to the 'Registration Estimator APP' provided in MATLAB. As per my understanding, in this step, the authors are trying to reversely deform the images so that the same blood vessels are aligned. In this way, the in-plane motions can be addressed, but the out-of-plane motions are not. Is there any estimation of the scale of the out-of-plane motion during diastole? How much does it affect the velocity measurement? What are the criteria for the image registration procedure to converge? How many iteration steps are set for the affine transformation estimation, and how many iteration steps are set for the non-rigid estimation? How many levels of the image pyramid are used, is there any smoothing of the non-rigid displacement field?
4. In lines 332-334, the correlation of the PSF with the image would further blur the original image. In this way, the closely located bubbles, e.g., where they are separated by less than half of the FWHM of the PSF, may not be sufficiently separated and the resulting location of the bubble might be wrong (since the authors are claiming a spatial resolution of $\sim 60\mu\text{m}$, which is much smaller than the wavelength). Considering the size of the PSF, it might cause negligible errors in the reconstruction of microvessels with a diameter of a few hundred microns or those that are separated by a few hundred microns. Can authors provide evaluations of bubble localization uncertainties for the currently used methods?
5. In lines 337-338, the motion-model-based methods and graph-based data association are widely used methods in many applications associated with radar multi-object tracking and object tracking velocimetry, etc. The authors may also acknowledge a few works in these fields.
6. In lines 350-352, why did the authors plot trajectories with a width that is a quarter of the wavelength, instead of the bubble's diameter (here it is smaller than the pixel size, so why the width of trajectories is not 1 pixel)? What is the basis of this selection? The Gaussian smoothing will affect the measured size of the blood vessels, and considering the scale of the blood vessel and the wavelength, the error may not be negligible. It would affect the velocity measurement in the same way. Can authors provide a detailed evaluation of this issue?
7. In figure 3(d) compare with the CT image in a, the vessel on the right side of LV was missing in US image.
8. In figure 4(b), The author should explain why there was fewer vessels on the left and right side of the heart compare with the top and bottom part.
9. In line 183, for vessel flow speed classification, did the author consider the acceleration rate of the moving bubble to exclude the incorrect pairing of MB?
10. In figure 5, the author should explain the color change in the zoom image compare with the original image.
11. For fig 6, the authors should provide more details in calculating all the vessel diameter. How did you locate and separate all the vessels in the image?
12. How would body fat affect the imaging results?
13. How would patients' breathing affect the results?
14. Sample size is too small for this study.

Reviewer #2 (Report for the authors (Required)):

Yan et al. report convincing results on the translation of ultrasound localization microscopy in cardiac

patients, enabling the transthoracic detection of intramural vessels in the anterior and posterior myocardial walls. Their methodology which relies on a clinical cardiac phased array probe, a nonlinear pulse sequence and post-processing at the radiofrequency and beamformed data levels represents an advance in the field. Demonstrating that ultrasound can image intramural coronary vessels has broad implications as it is the first technology that can chart this vascular territory noninvasively in cardiac patients.

Major comments:

1. While the patient data reported by Yan et al. is of significant importance, data analysis does not adhere to best practice in ultrasound localization microscopy and should be improved. Authors should assess in vivo resolution using at least Fourier ring correlation curves (report FRC curves and not an FRC value as you do - see Heiles et al. Nature BME 2022) and quantitative assessment of velocity profiles across arteriole sections (see Demene et al. Nature BME 2021, figure 3e-f). Bolus saturation curves, corresponding to the image area covered by MB detections as a function of time should also be reported (see Hingot et al. IEEE TMI 2021).
2. A major limitation of ultrasound localization microscopy is that the absence of a vessel can either be due to a perfusion defect, which has diagnostic value, or to an incomplete mapping of vascular networks by microbubbles, which is a technical limitation. Given the relatively short acquisition times reported here (10 cardiac cycles max), you cannot generate complete maps of the coronary vasculature. How will you dissociate impaired myocardial microcirculation from incomplete mapping of vessels due to short acquisition times? Your images are still far from revealing the complete coronary vasculature. This should be discussed.
3. I am puzzled by flow profiles reported in Figure 5a. I would have expected higher flow velocities in the epicardial coronary segments than in the penetrating arterioles. Can you provide the upper and lower flow velocity limits that can be detected with your current pipeline, and discuss the impact of vessel orientation on your results?
3. Data processing steps to generate your supplementary video are not described. What period of the cardiac cycle are you showing precisely? Is the movie displaying consecutive diastoles or are you looping over a single averaged diastole? You have to ensure reproducibility of your results.

Minor comments:

1. Remove super-resolution from the title, ultrasound localization microscopy is enough and more accurate.
2. Remove SRUS from the text and stick to ULM.
3. Line 47: this sentence is wrong. ULM is a class of super-resolution methods among others. Please correct.
4. Line 96: describe data processing performed on the RF data in supplementary figure 1 or another supplementary figure, this is interesting.
5. Line 102: can you quantify and report myocardial motion in your ex vivo and in vivo datasets? You could use tissue Doppler imaging for example.
6. Figure 2: why is the apical vasculature not detected (Figure 2e)? Where was the cross section of the heart taken, at the basal, mid or apical level (Figure 2c)?
7. Lines 202-203: the claim about improved management of cardiac patients is not supported by your data.
8. The benefit of nonlinear Doppler is not immediately clear to me, please expand. You could refer to efforts to design more specific ultrasound pulse sequences.
9. Line 233, cite Demeulenaere et al. JACC Imaging 2022.
10. Line 265. Beside the center frequency, can you indicate the transmit frequency you used for your AM sequence?
11. Report the transmitted pressure as a function of depth in a supplementary figure.
12. Report estimated PSFs as a function of depth in a supplementary figure.

Reviewer #3 (Report for the authors (Required)):

The paper by Yan et al. reports of super-resolution ultrasound techniques in order to visualize microvascular cardiac disease in patients with a simple echocardiogram. The data presented are very compelling and demonstrate the potential of this technique to be used clinically for the early detection of heart disease. One shortcoming that could be addressed by the authors is to pinpoint the methodology that made it this imaging possible. This is because all the techniques presented such as MoCo, SVD etc. have previously been implemented by the authors and others so it is not clear whether a specific breakthrough needed to be accomplished. Was it the fuzzy initialization? The AM sequence? If so, why? Some important methodological details are lacking which would aid further in the assessment of the performance of the technique. The transthoracic implementation is exciting but not clear why it was possible in the first place. Along those lines, results before and after different implementations would be helpful. Finally, some syntax and grammatical errors appear occasionally throughout which should be easily fixed.

1. Line 38 – ‘CMD is also associated....’ Please provide references for the statements made here.
2. Line 74- Please be more specific regarding the type of ‘impaired left ventricular function’ that the patients in this study were diagnosed with.
3. Line 87 – How was the AM sequence used here to separate the bubble signal? Please specify all parameters and rationale used and whether this was a key aspect of the technique.
4. Line 89- please specify the concentration, size distribution and composition of the bubbles used here. Was the dosage used here the clinical dose?
5. Lines 96 and 106: Please provide all methodological details for the AM sequence and the SVD filtering, respectively.
6. Lines 116- how was the Kalman motion model used here? Please provide all important methodological details.
7. Fig. 2e. There seems to be an angle dependence of the technique in the Langerdorff model, the lateral walls are very dark. Please indicate whether this is a limitation of the technique or it's in vitro implementation.
8. Line 130-‘myocardium surround the left ventricle...’ Please correct for syntax/grammar.
9. Fig. 3d. There seem to be dark regions in the image, similar to the in vitro case. Is this due to the divergent beamforming sequence? Please provide an explanation on the potential angle dependence.
10. Line 158-Please indicate the epicardium and endocardium on the figure.
11. Figure 3- Please show images before and after MoCo to better evaluate the effect of the technique. What was the range of motion estimated in vivo? How did it compare to in vitro?
12. Line 180- How were these vessels validated? Please indicate how ICM could be evaluated using this technique.
13. Line 183: What is the significance of quantifying coronary flow velocity? Could this aid in identification of ischemia and/or infarction or treatment thereof?
14. Figure 4. Similar dark regions in the lateral wall in (b). Please explain.
15. Line 200: Please quantify the improvement over the diffraction limit.
16. Line 223: Please indicate whether the clinical findings were in agreement with the clinical diagnosis and why.

Tue 17 Oct 2023

Decision on Article nBME-23-0750A

Dear Prof Tang,

Thank you for your revised manuscript, "Transthoracic ultrasound localization microscopy of myocardial vasculature in patients", which has been seen by the original reviewers. In their reports, which you will find at the end of this message, you will see that the reviewers acknowledge the improvements to the work and raise a few additional technical criticisms that we hope you will be able to address.

As before, when you are ready to resubmit your manuscript, please upload the revised files, a point-by-point rebuttal to the comments from all reviewers, the reporting summary, and a cover letter that explains the main improvements included in the revision and responds to any points highlighted in this decision.

As a reminder, please follow the following recommendations:

- * Clearly highlight any amendments to the text and figures to help the reviewers and editors find and understand the changes (yet keep in mind that excessive marking can hinder readability).
- * If you and your co-authors disagree with a criticism, provide the arguments to the reviewer (optionally, indicate the relevant points in the cover letter).
- * If a criticism or suggestion is not addressed, please indicate so in the rebuttal to the reviewer comments and explain the reason(s).
- * Consider including responses to any criticisms raised by more than one reviewer at the beginning of the rebuttal, in a section addressed to all reviewers.
- * The rebuttal should include the reviewer comments in point-by-point format (please note that we provide all reviewers will the reports as they appear at the end of this message).
- * Provide the rebuttal to the reviewer comments and the cover letter as separate files.

We hope that you will be able to resubmit the manuscript within 4 weeks from the receipt of this message. If this is the case, you will be protected against potential scooping. Otherwise, we will be happy to consider a revised manuscript as long as the significance of the work is not compromised by work published elsewhere or accepted for publication at *Nature Biomedical Engineering*.

We look forward to receive a further revised version of the work. Please do not hesitate to contact me should you have any questions.

Best wishes,

Liqian

Dr Liqian Wang
Associate Editor, Nature Biomedical Engineering

Reviewer #1 (Report for the authors (Required)):

The author has answered most questions and revised the manuscript. However, ULM technique has been proposed over a decade with both technique and application advances, including animal brain imaging (Errico et al., Nature 2015), 4D mapping (Rabut et al., Nature Methods, 2019), functional ULM (Renaudin et al., Nature Methods, 2022) and patient study (Demene et al., Nature BME, 2021). It seems that this study only shifts existing ULM technique to another application – myocardial vasculature. In addition, the clinical

benefit of using ULM is not clear since no extra information was provided in this study. Overall, the advances in both clinical impact and ULM methodology are questionable.

1. One key advantage of ULM technique is to provide much higher resolution. However, the reconstructed ULM image quality in this study is poor. For instance, the quantified resolution is ranging from 150 to 250 μm which is very close to the half-wavelength of the transmission pulse (320 μm for 2.4 MHz transducer). Therefore, the advantages of using ULM are questionable and should be carefully addressed. Small resolution improvements could be realized by other approaches instead of time-consuming ULM processing.
2. The flow speed of the myocardial vasculature has a huge variation during the cardiac cycle. Does the presented ULM technique have the capability to cover such a large range? If so, please provide the calibration results and its comparison with the ground truth (i.e., use optical image as standard). In addition, the effective frame rate is only 305 Hz with limited acquisition time, how does the author ensure the accuracy of your ULM reconstruction? The images presented in this study are averaged over time, right? It will lose temporal information which is a drawback of ULM but important for heart observation.
3. MB concentration is an important parameter in ULM reconstruction. How does the author control the accuracy of MB localization and tracking under a high MB concentration as well as a high flow range? Please specific it in detail. The author should also explain why there was limited detectable vessels in the images which is contradiction to the heart structure (plenty of vessels). In addition, results are presented for the diastolic phase, why not for a full cardiac cycle?
4. Apart from other organs – liver, kidney, brain, motion artifacts of heart could significantly downgrade the ULM quality even hold the breathing. Although authors implemented motion registration algorithms to compensate its morphological changes, the rapid heart volume alternation will largely affect the MB distribution, resulting in uncertain trajectory, out-of-plane movement and so forth. How does the author address this issue?
5. If this study is designed for a new application – cardiac disease, please highlight the significance of quantifying small vessels. According to the presented ULM image, the resolution improvement is not enough (from 320 μm in theoretical to ~ 250 μm in measurement). In addition, the advantages over conventional imaging such as CTCA, functional MRI is not explained in this study. Moreover, many methodological details have been proposed by the author's group and others. The reviewer is not clear about the technique advance. Is there any novel approach applied to make this cardiac vasculature feasible? Please highlights this in detail.

Reviewer #2 (Report for the authors (Required)):

Comments to the authors:

1. Please include FRC curves in one of the main figures of the manuscript rather than in supplementary, this is proper metric to report resolution. Even though you "only" reach $\lambda/2$, you are still reporting the highest resolution ultrasound images of the coronary vasculature in patients to date. Pay attention in the text, it is FRC and not FSC, please correct.
2. Can you add to the text a comparison of your FRC resolution and B-mode resolution? This is what matters in the end and it will compare more favorably that the wavelength.
3. Given the reported FRC resolution, I would not claim in the abstract that you detect vessels beyond the resolution limit but at the resolution limit ($\lambda/2$). Please correct.
4. Figure 5, please perform a bin analysis of the velocity profiles as in Errico et al. Nature, 2015 figure 2E.
5. I still have a question regarding the data processing steps to generate your supplementary video. What do you mean by "7) Microcirculation demonstration"? Please describe these processing steps in the text.
6. Remove "SRUS/ULM" from the text and simply use ULM throughout the manuscript. Ultrasound localization microscopy (ULM) is a subclass of vascular super resolution ultrasound (SRUS) methods. Your

sentence line 48 is still not accurate. Please rephrase as: "Ultrasound localization microscopy (ULM), a class of super-resolution ultrasound (SRUS) methods, can image deep microvasculature [...]".

Reviewer #3 (Report for the authors (Required)):

The authors did an excellent job responding to the comments with new figures, new videos and new supplementary data. Some of the details of the method regarding improvements still remain unclear especially regarding the details of the AM approach and the MoCo method in vivo so it is highly recommended that the authors increase the rigor by providing parameters that they believed increased the image quality reported.

Tue 19 Dec 2023

Decision on Article nBME-23-0750B

Dear Prof Tang,

Thank you for your revised manuscript, "Transthoracic ultrasound localization microscopy of myocardial vasculature in patients". Having consulted with the original reviewers (whose comments you will find at the end of this message), I am pleased to write that we shall be happy to publish the manuscript in *Nature Biomedical Engineering*.

We will be performing detailed checks on your manuscript, and in due course will send you a checklist detailing our editorial and formatting requirements (Given the extremally high workloads I'm bearing currently, this process would probably take for one month). You will need to follow these instructions before you upload the final manuscript files.

Best wishes,

Liqian

Dr Liqian Wang
Associate Editor, Nature Biomedical Engineering

Reviewer #1 (Report for the authors (Required)):

The authors have answered and revised most questions from reviewers. It was suggested to accept it for publication.

Reviewer #2 (Report for the authors (Required)):

I am satisfied with the answers provided by the authors and have no further comments.

Reviewer #3 (Report for the authors (Required)):

The authors responded adequately to my comments.

Rebuttal 1

RE: Decision on Article nBME-23-0750

Dear Editor and Reviewers,

The authors are grateful for the advice and comments. We have listed our responses and revisions in this rebuttal, which we hope would address the concerns and questions raised by the reviewers.

Revised texts are highlighted by **yellow colour** in the paper. The main corrections in the paper and the answers to the reviewer's comments are given in two sections. The first section is suggested by the editor to address the comments raised by more than one reviewer. Responses to each remained comments can be found in the second section.

Section 1: Responses to Comments Raised by More Than One Reviewer

C1. Motion estimation and correction.

Comments from reviewers:

1) Reviewer#1, Part of Comment 3. In this way, the in-plane motions can be addressed, but the out-of-plane motions are not. Is there any estimation of the scale of the out-of-plane motion during diastole?

Response: we acknowledged that the out-of-plane motion is difficult to estimate, due to the inherent limitation of 2D ultrasound, and thus ultrafast 3D cardiac ultrasound would be the way forward for full 3D motion estimation and correction. To answer the reviewer's question, we think this may be roughly estimated by imaging at different views. The maximal in plane deformation on short- and long-axis views in our human data, as estimated by the motion estimation method, is around 1.7 and 1.4 mm respectively during a diastole phase, visual demonstration of which can be found in Supplementary Video 3 and 4. As these two views are approximately orthogonal to each other, the in-plane motion in one view could be indicative of the out-of-plane motion in the other view. Therefore, we believe the maximal out of plane motion during diastole in this case is approximately 1.7 mm. This motion is expected to be larger during systole.

2) Reviewer#2, Part of Comment 5. Line 102: can you quantify and report myocardial motion in your ex vivo and in vivo datasets?

Response: According to the motion estimated by the motion correction algorithm, the ex vivo pig heart during a whole cycle and the in vivo human heart in diastole phases have similar magnitude of myocardial motions, maximum of which was between 1.4 and 1.7 mm. Visual demonstration of is in Supplementary Video 2 and 3. Please note that the pig heart at the time of measurement was not beating normally, hence the small myocardial motion over the full cardiac cycle, likely due to ischemia as further discussed in Comment C2.

3) Reviewer#3, Part of Comment 11. What was the range of motion estimated in vivo? How did it compare to in vitro?

Response: See our responses above.

Added Content:

We provide three additional Supplementary Videos to demonstrate the motions in data and the effectiveness of used motion correction algorithms.

Supplementary Video 2: Short-axis view B-mode frames of Porcine 1 before (left) and after (right) motion correction. Dynamic range: 50 dB.

Supplementary Video 3: Short-axis view B-mode frames in one diastole phase of Patient 1 before (left) and after (right) motion correction. Dynamic range: 50 dB.

Supplementary Video 4: Long-axis view B-mode frames in one diastole phase of Patient 1 before (left) and after (right) motion correction. Dynamic range: 50 dB.

Below sentences have been added in the result section.

“The detected in-plane tissue deformation on the short-axis view between two frames during one cardiac cycle of a porcine heart and on the two views during the diastole phase of a human heart is similar and ranges between 1.4 and 1.7mm. Comparisons of B-mode frames before and after motion correction are demonstrated in Supplementary Video 2, 3 and 4 for the three views respectively. Effectiveness of this two-level motion correction strategy is also demonstrated when comparing Power Doppler image obtained with and without motion correction, as shown in Fig. F2 of the Supplementary Figures.”

C2. Dark regions in SR density maps of Langendorff model.

Comments from reviewers:

- 1) Reviewer#2, Part of Comment 6. Figure 2: why is the apical vasculature not detected (Figure 2e)?
- 2) Reviewer#3, Comment 7. Fig. 2e. There seems to be an angle dependence of the technique in the Langerdorff model, the lateral walls are very dark. Please indicate whether this is a limitation of the technique or it's in vitro implementation.

Responses to both comments:

It is possible that the apical vessels in Figure 2e were damaged or blocked, for example due to inadvertent entry of air. Vessels above and below that dark area in Figure 2e are visible; Vessels at the same position in Figure 2c were also visible. Therefore, any abnormality is local to the apex only. We carried out an additional ex vivo experiment and vessels at the same position (Lateral: 20 to 40mm, Depth: 30 to 50 mm) in the reconstructed image are also visible, as shown Fig. F8. Therefore, while the imaging sensitivity may decrease towards the lateral boundary, we believe lack of blood flow in that particular specimen is the main reason for the dark regions in Fig. 2.

Added Content:

We carried out an additional ex vivo experiment with the probe targeted at the apical and the reconstructed SR map is shown below. The vessels in the lateral walls are as visible as top and bottom parts.

Fig. F7 | ULM image reconstructed from data acquired from Langendorff ex vivo porcine heart 2.

We therefore believe the dark areas in the ex vivo heart 1 is due to vascular damage/blockage and added below sentence in the Result section to clarify the question.

“There is a dark region in the lateral area of Fig. 2e while some vasculature below and above that area is still visible. Furthermore, the vasculature in the same position in the second ex vivo heart, as shown in Fig. F7 of the Supplementary Figures, is clearly visible. Therefore, it is likely the apical vasculature in that area in ex vivo porcine heart 1 was damaged or blocked.”

C3. Few vessels at the lateral sides of the reconstructed SR density maps of patient 2.

Comments from reviewers:

- 1) Reviewer#1, Comment 8. In figure 4(b), The author should explain why there was fewer vessels on the left and right side of the heart compare with the top and bottom part.
- 2) Reviewer#3, 14. Figure 4. Similar dark regions in the lateral wall in (b). Please explain.

Responses to both comments:

We believe the fewer vessels in the lateral walls could be due to side lobes. As demonstrated in Fig.F3 showing the power Doppler images for this patient, side lobes of the large and bright chambers can reduce the contrast at the depth of the chamber. The CV beamforming can reduce sidelobes but can also reduce the bubble signals at these locations.

Added Content:

We added below content into the Supplementary Figures to demonstrate the effect of side lobes of chambers on contrast.

“

Fig. F3 | Power Doppler images obtained by beamforming data acquired from Patient 2 with DAS (a) and CV (b) beamformer. Strong contrast signals from within the ventricle chamber can generate significant side lobe artefacts on the side wall, which can be seen from Fig. F2b and the Fig. F3a. Side lobes of chamber can significantly affect the detection of MBs. The CV beamformer can help reduce side lobes but can also reduce the intensity of microbubble signals at the side lobe area. This may be the reason why less vessels are seen on the side wall of patient two. More advanced side lobe reduction algorithms should be explored.”

Below sentence has been added into the Result section.

“There was less reconstructed vasculature at lateral sides than the top and bottom parts in the Fig. 4b. It may be because of the strong side lobes of the large and bright chamber. More vessels in the side wall are visible in Fig. 2c, Fig. 3g and Fig. F7 where the chambers are smaller.”

Below sentence has been added into the Discussion section.

“Side lobes of chambers can also reduce contrast of MB images, as shown in Fig. F2b and Fig. F3a of the Supplementary Figures, and thus reduce sensitivity of localisation near the chamber. Image reconstruction method that can further reduce side lobes, especially those of chambers, is worth exploring.”

C4. Implementation of CEUS image reconstruction

Comments from reviewers:

1) Reviewer#2, Comment 4. Line 96: describe data processing performed on the RF data in supplementary figure 1 or another supplementary figure, this is interesting.

Response: A diagram for reconstructing images from the RF data has been added into the Supplementary Figure F5.

2) Reviewer#3, Comment 5. Line 96: Please provide all methodological details for the AM sequence.

Response: Line 96 is about image reconstruction, and we have now added a diagram in the Supplementary Figure F5 to present details. For details about AM sequence please refer to our response to your Comment 3.

Added Content:

We add below figure into the Supplementary material to illustrate the data processing performed on RF data.

Fig. F5 | Processing for CEUS image. **a**, AM was done to remove linear tissue signals. **b**, moving average subtraction was done to remove remained nonlinear tissue signals. **c**, image in polar coordinate was generated for each angle and each channel. **d**, images were summed across channels corresponding to each angle. **e**, MB shifts during steering angles were estimated by the Doppler-based method. **f**, MB motions in each channel images were corrected by the phase shifts estimated from the corresponding steering angles. **g**, CEUS images were reconstructed with the coherence to variance (CV) beamforming, where the variance was calculated across channels and steering angles for each pixel.

Section 2: Responses to Each Reviewer.

Reviewer #1:

In this work, the authors have demonstrated the feasibility of using transthoracic ULM for investigating the myocardial microvascular hemodynamics on one large animal model and two patients, but only during the diastolic phases.

The topic of imaging myocardial microvessels is interesting and it definitely opens up more feasibilities for a broader range of clinical applications of ULM. Addressing the motion correction issues of imaging the heart sections with dramatic 3D motion is challenging without volumetric imaging, yet the authors have set a starting point by using available image registration techniques to correct motions during diastolic phases. This manuscript demonstrates the use of super-resolution ultrasound images for cardiovascular vascular imaging. One ex vivo experiment was performed with a porcine heart and two patient was imaged and the authors has demonstrated the intra-cardiac-cycle and inter-cardiac-cycle tissue motion correction to overcome the Influence of heartbeat.

We thank the reviewer for the encouraging comments.

However, the overall image quality was poor comparing with other SRUS image. There are some concerns as follow:

1. Instead of tracking the bubbles, which takes a long time for data processing, what does the power Doppler with contrast agents look like for the myocardial microvessels?

Response: We have now included below power Doppler image in Supplementary Figures for comparison as suggested. By comparing below image with the Fig. 3g in the main text, SRUS technique can present the vascular structure with much higher resolution than the power Doppler, and generate flow speed and direction maps presented in Fig. 5. It should be noted that the motion correction step, which is also used in generating the power Doppler images, take much longer, around 2 s per frame varying with the number of iterations, than other data processing steps used in SRUS including tracking, around 0.5 s varying with number of MBs.

Fig. F2 | Power Doppler images of CEUS sequence acquired from the short-axis view of Patient 1 and reconstructed with different methods.

2. In lines 304-305, why haven't the authors used EKG signals for identifying the cardiac cycles?
Response: We used a research platform (Verasonics) for ultrasound data acquisition and while we have a separate ECG acquisition system which can be connected to the system, we have not yet obtained ethics approval to use it on patients. As our goal is to find the most similar images to register and accumulate, one could argue that gating cardiac cycle with images could be as suitable for motion correction.

3. In lines 309-310, this two-stage image registration method seems very similar to the 'Registration Estimator APP' provided in MATLAB. As per my understanding, in this step, the authors are trying to reversely deform the images so that the same blood vessels are aligned. In this way, the in-plane motions can be addressed, but the out-of-plane motions are not. Is there any estimation of the scale of the out-of-plane motion during diastole? How much does it affect the velocity measurement? What are the criteria for the image registration procedure to converge? How many iteration steps are set for the affine transformation estimation, and how many iteration steps are set for the non-rigid estimation? How many levels of the image pyramid are used, is there any smoothing of the non-rigid displacement field?

Response: The non-rigid registration in the APP uses the demons method, proposed in Thirion, Medical Image Analysis, 1998. We clarified the used image registration technique in the reporting summary.

"We used existing B-spline free-form registration method. A CPU code version can be found in Matlab Central File Exchange (<https://www.mathworks.com/matlabcentral/fileexchange/20057-b-spline-grid-image-and-point-based-registration>). We developed a CUDA version for faster computation."

The out-of-plane motion during diastole is estimated to range between 1.4 and 1.7mm. Details of this estimation can be found in our response to comment C1.

In the current tracking algorithm used for 2D ultrasound, MBs were tracked by their lateral and depth positions. Out-of-plane motions change the elevational positions of MBs and change MB locations in the imaging plane when delay related to elevational position is not considered in image reconstruction. Provided that elevational thickness is much smaller than the imaging depth, out-of-plane motion should not affect the in-plane velocity measurement until the MBs cannot be detected and tracked anymore.

We tried two pyramid levels but found no obvious improvement in our case. To save computation and parameters to be tuned, we used a single pyramid level. We further tried a few parameters and below content has been added into the method section to state the used parameters.

"Sum of square pixel intensity difference was used as image difference for both the affine and non-rigid image registration. The grid of B-spline control points was set with a spacing of 16 (for ex vivo) or 32 (for in vivo) pixels along both the lateral and depth. Unrealistic non-rigid deformation might be generated after registration when speckle patterns changed with out-of-plane deformation and wave interference. The thin-plate spline constraint was used as a penalty to restrict the distortion [temp]. Affine and non-rigid registration was solved by the Levenberg–Marquardt and the steepest decent algorithms respectively. The maximum iteration steps for affine and non-rigid transformation were

both set as 500. Each stage of the optimisation would stop earlier if the image difference between two adjacent iterations changed less than 0.001% for 20 times. The first stage optimisation would also stop earlier when the difference between transformation parameters in two iterations was less than 1×10^{-4} , e.g., around 4.4×10^{-6} wavelength for translation along depth.

[temp] Rueckert, D. et al. Nonrigid registration using free-form deformations: application to breast MR images. IEEE Trans. Med. Imaging 18, 712–721 (1999)

4. In lines 332-334, the correlation of the PSF with the image would further blur the original image. In this way, the closely located bubbles, e.g., where they are separated by less than half of the FWHM of the PSF, may not be sufficiently separated and the resulting location of the bubble might be wrong (since the authors are claiming a spatial resolution of $\sim 60\mu\text{m}$, which is much smaller than the wavelength). Considering the size of the PSF, it might cause negligible errors in the reconstruction of microvessels with a diameter of a few hundred microns or those that are separated by a few hundred microns. Can authors provide evaluations of bubble localization uncertainties for the currently used methods?

Response: We fully agree that the correlation calculation can further blur the original image and reduce the possibility of separating closely located bubbles from neighbouring vessels. Separating closely located bubbles is a common challenge in super localisation, and the correlation method makes it more challenging, and relies more on bubbles sparsely distributed in spatial and temporal domain. We chose the cross-correlation as we found it to be robust to SNR, although future work could consider other methods such as Gaussian model fitting.

We evaluated the bubble localisation uncertainties of the system by imaging a wire target under water. The values of localised uncertainties are listed in the Table S1.

Finally, we have revised our image resolution evaluation using Fourier Ring analysis with a different data splitting method following a previous literature. The details of the revised resolution assessment can be seen on page 15, reviewer 2, comment 1.

Table S1 | Localisation uncertainty measurements with in vitro experiments. A wire target was fixed under three different distances from the probe and imaged for 600 frames per depth using the same acquisition settings of the in vivo experiments. Localisation uncertainties were quantified by the variance of localised positions at each depth. Both transmission frequencies, i.e., 1.7 and 2.4 MHz, used in in vivo acquisition were tested.

Depth (mm)	1.7 MHz		2.4 MHz	
	Lateral uncertainty (um)	Axial uncertainty (um)	Lateral uncertainty (um)	Axial uncertainty (um)
30	1.94	1.23	7.66	9.09
80	4.80	2.48	3.29	3.02
120	9.40	1.84	15.54	1.17

We add below sentence into the Method Section.

“The localisation uncertainties of the system were measured no more than 16 μm by imaging a wire target at three different depths in water, as demonstrated in Table S1 of the Supplementary Table.”

We add below sentence into the discussion section.

“Normalised cross-correlation, smoothing CEUS images with PSFs, was chosen as the localisation method to deal with limited SNR in CEUS images but reduced the probability of separating closely located bubbles. This might cause nonnegligible errors in the reconstruction of microvessels with a diameter of a few hundred microns or those that are separated by a few hundred microns, and hence rely more on spatial sparsity of bubble signals.”

5. In lines 337-338, the motion-model-based methods and graph-based data association are widely used methods in many applications associated with radar multi-object tracking and object tracking velocimetry, etc. The authors may also acknowledge a few works in these fields.

Response: We added below citations to acknowledge motion model and multi-object tracking work in other fields.

[xx] Fortmann, Y. Bar-Shalom, and M. Scheffe, “Sonar tracking of multiple targets using joint probabilistic data association,” *IEEE J. Ocean. Eng.*, vol. 8, no. 3, pp. 173–184, Jul. 1983.

[xx] S. Oh, S. Russell, and S. Sastry, “Markov chain Monte Carlo data association for multi-target tracking,” *IEEE Trans. Autom. Control.*, vol. 54, no. 3, pp. 481–497, Mar. 2009.

[xx] D. Reid, “An algorithm for tracking multiple targets,” *IEEE Trans. Autom. Control*, vol. 24, no. 6, pp. 843–854, Dec. 1979.

We added below citations to acknowledge the graph-based assignment algorithm used in our study.

[xx] Hichcock, F.L., 1941. The distribution of a product from several sources to numerous localities. *J. Math. Phys.* 20, 224.

[xx] Sbalzarini, I. F. & Koumoutsakos, P. Feature point tracking and trajectory analysis for video imaging in cell biology. *J. Struct. Biol.* **151**, 182–195 (2005).

Any suggestions of other relevant literatures would also be gratefully received.

6. In lines 350-352, why did the authors plot trajectories with a width that is a quarter of the wavelength, instead of the bubble’s diameter (here it is smaller than the pixel size, so why the width of trajectories is not 1 pixel)? What is the basis of this selection? The Gaussian smoothing will affect the measured size of the blood vessels, and considering the scale of the blood vessel and the wavelength, the error may not be negligible. It would affect the velocity measurement in the same way. Can authors provide a detailed evaluation of this issue?

Response: The reviewer raised a very important point of how SRUS images should be plotted and be visualised for our eyes, given the typical large field of view in in vivo imaging and the small microvessels, particularly when only a small sample of the vascular network is reconstructed due to the limited acquisition time, as demonstrated in the added Fig. F14 in the Supplementary Figure.

Fig. F14| Density map of the short-axis view of Patient 1 using a single pixel size, 13.5 μm , which is approximately 1/60 of wavelength, without smoothing and the magnified image in the blue box. Visually, vessels were far from being fully saturated by the trajectories due to the limited acquisition time. The selection of a quarter of the wavelength was aimed to improve the visibility of the microvessels in the image.

We fully acknowledge that smoothing can affect the measured size of blood vessels, as larger Gaussian can increase vessel size measurements, or merge closely separated trajectories into larger vessels. A trade-off is required when choosing the Gaussian size, especially when saturation is limited as multiple tracks in a large vessel could be mistaken as multiple small vessels.

The velocity measurement does not seem to be affected by smoothing as shown below.

Below table shows the average velocity and the standard deviations calculated from the short-axis view of patients using different sizes of smoothing. The velocity was measured by pixel values on the SR velocity map, pixel value of which was obtained by the division of disk-smoothed speed to the disk-smoothed density map, instead of implementing a simple disk filtering on unsmoothed speed map. With this implementation, velocity measurement does not change significantly with the smoothing size.

Table| Velocities, mean and standard deviations, calculated from the maps plotted with difference sizes of disk smoothing. λ is the wavelength.

diameter for disk smoothing		$\frac{\lambda}{2}$	$\frac{\lambda}{4}$	$\frac{\lambda}{8}$
Velocity(mm/s)	Patient 1	54.9 ± 25.6	55.4 ± 24.5	55.7 ± 25.8
	Patient 2	54.9 ± 23.1	55.8 ± 25.0	56.6 ± 26.2

Below sentence in the Method Section has been revised to clarify the selection of smoothing.

The localisation density map was plotted by accumulating trajectories with a width defined by a 2D Gaussian, using MATLAB function 'imgaussfilt', whose FWHM was set as a quarter of ultrasound wavelength at the transmitted centre frequency to enhance image saturation due to the limited

acquisition time. One density map plotted directly using the pixel size is presented in Fig. F14 in the Supplementary Figures.”

Below sentence has been added into the discussion to clarify the limitation in diameter measurement.

“The vessel size measurements can be affected by the size of the 2D Gaussian used for plotting super-resolution images. When a small size is used, multiple tracks in a large vessel could be mistaken as multiple small vessels when the saturation is limited. A larger Gaussian used in plotting density maps can help reconstruct large vessels from separated trajectories but will increase measured vessel size and can also merge separate small vessels that are close to each other. A trade-off is required, particularly when the vascular signal saturation is low.”

Below sentence in Method Section has been revised to clarify the implementation of smoothing for velocity map.

“Flow speed map was calculated by division of the disk-smoothed accumulated speed to the disk-smoothed accumulated density, instead of directly applying smoothing on an unsmoothed speed map, to reduce the effect of smoothing kernel size on velocity measurement.”

7. In figure 3(d) compare with the CT image in a, the vessel on the right side of LV was missing in US image.

Response: Below sentence has been revised in the main text for explanation.

“It should be noted that the CT images correspond to a slab with a constant thickness of 10 mm, while the SRUS has a much smaller slab thickness (a typical cardiac probe has a slab thickness of one to a few wavelengths varying in depth, less than 5 mm for the used probe). Due to slab thickness difference or the low SNR of ultrasound below the chamber, the vessel on the right side of LV is presented in CT image Fig. 3a but missed in the ULM image Fig. 3d.”

8. In figure 4(b), The author should explain why there was fewer vessels on the left and right side of the heart compare with the top and bottom part.

Response: Please find the response to Comment C3.

9. In line 183, for vessel flow speed classification, did the author consider the acceleration rate of the moving bubble to exclude the incorrect pairing of MB?

Response: The acceleration of MB movements was considered during MB paring. The acceleration affects the model prediction error when assuming constant velocity in the linear Kalman motion model. Although we had no prior knowledge about the prediction error, we estimated the variance of prediction error from the data by the supplementary method.

10. In figure 5, the author should explain the color change I the zoom image compare with the original image.

Response: The original and zoom images are flow velocity and direction maps respectively. We added labels on the top of images and the velocity profile as suggested by the second reviewer.

Fig. 5 | Ex vivo porcine heart and in vivo patient short-axis view of flow speed, direction, flow speed profile. a, d, the flow speed maps, corresponding to the SR density map in Fig. 2 and 3. b, c, SR flow direction map of the zoomed-in regions inside the white boxed in a and b. e, f the flow speed profiles at the positions indicated by the red lines in b and c, which were generated by speeds on segments averaged across 0.5 mm width along the vessels.

11. For fig 6, the authors should provide more details in calculating all the vessel diameter. How did you locate and separate all the vessels in the image?

Response: The sentences about metric calculation were revised as below.

“Vessel diameters were measured with vessel centre lines and boundaries without segmenting vessels. The centre lines of vessels were detected on the binarized density map by Matlab “bwmorph” function with ‘skel’ property set to ‘inf’. ‘bwdist’ function was applied to the inversed binarized density map to obtain the distance between the centre line and the nearest boundary of the vessel. The vessel diameter was defined by doubling the distance corresponding to each pixel on the centre lines.”

12. How would body fat affect the imaging results?

Response: We believe body fat can significantly affect imaging results, by reducing signal to noise ratio and generating aberration that may reduce the sensitivity of bubble detection. We had discussed about phase aberration in our last submission. Below sentences have been added into the discussion about optimisation in acquisition for improving MB detection.

“Optimisation for improving MB detection can also be done in acquisition. For example, transmitted pressure might need to be higher for the patients with higher BMI, i.e., thicker body fat, which can make heart further away from the probe.”

13. How would patients breathing affect the results?

Response: Patients’ breathing would introduce more out-of-plane motion that is difficult to be corrected with 2D ultrasound. Besides, the breathing would affect the imaging slices of myocardium with the limited acoustic window between ribs. Therefore breath holding is a common requirement for clinical echocardiography. We have discussed the breathing problem in the paragraphs about acquisition and the usage of 3D ultrasound. Below sentence has been revised in Discussion section.

“Patients were asked to adjust their breath to get appropriate imaging views of heart according to clinician experience in conventional echocardiograph examination and then hold breath to reduce out-of-plane motion, which is important for 2D ultrasound acquisition and restrict the acquisition time.”

14. Sample size is too small for this study.

Response: We fully acknowledge the preliminary nature of this study that has very small sample size. Our primary aim is to initially demonstrate that SRUS of myocardial vasculature is possible. We have since increased our sample size, with one ex vivo data set from a second Langendorff porcine heart and in-human data set from two additional patients. The reconstructed SR density images were included in the Supplementary Figures.

Fig. F7 | SR image reconstructed from data acquired from Langendorff ex vivo porcine heart 2.

Fig. F8 | *In vivo* 3D scan and SRUS images for Patient 3 and 4. **a**, the short-axis view in the CTCA scan that corresponds to the ultrasound imaging plane. White arrows point out similar structures visible in both modalities. **b**, the SRUS density map for myocardial microvasculature. **c**, the zoomed-in SRUS density map for the region in **b**. **e**, the short-axis view in the Cardiac MRI scan that corresponds to the ultrasound imaging plane. **f**, the SRUS density map for myocardial microvasculature. **g**, the zoomed-in SRUS density map for the region in **f**. Note that the bright reflection spot in the CTCA of Patient 3 was generated from the lead of the implantable cardioverter defibrillator, which impacts the imaging of CTCA. Patient 4 did not have clinical indication for a CT scan, so only MRI scan was conducted. The Cardiac MRI with perfusion was not acquired at the same slice with the ultrasound. One slice of 3D Cardiac MRI sequence, suppressing signal from flowing blood and maintaining high signal in the surrounding stationary tissues, is used here to illustrate the myocardium.

Reviewer #2

Yan et al. report convincing results on the translation of ultrasound localization microscopy in cardiac patients, enabling the transthoracic detection of intramural vessels in the anterior and posterior myocardial walls. Their methodology which relies on a clinical cardiac phased array probe, a nonlinear pulse sequence and post-processing at the radiofrequency and beamformed data levels represents an advance in the field. Demonstrating that ultrasound can image intramural coronary vessels has broad implications as it is the first technology that can chart this vascular territory noninvasively in cardiac patients.

We thank the reviewer for the encouraging comments.

Major comments:

1. While the patient data reported by Yan et al. is of significant importance, data analysis does not adhere to best practice in ultrasound localization microscopy and should be improved. Authors should assess in vivo resolution using at least Fourier ring correlation curves (report FRC curves and not an FRC value as you do – see Heiles et al. Nature BME 2022) and quantitative assessment of velocity profiles across arteriole sections (see Demene et al. Nature BME 2021, figure 3e-f). Bolus saturation curves, corresponding to the image area covered by MB detections as a function of time should also be reported (see Hingot et al. IEEE TMI 2021).

Response: We have generated the Fourier ring correlation curves and the velocity profiles across vascular sections, and saturation curves following the references as suggested (shown below). Please note that the revised resolution number is different from our previous one, as we were randomly splitting MBs' pairings between frames into two parts, which could bring additional dependence between the two parts if the pairings belong to the same MB. In this revised submission, we follow the method described in the paper [Hingot et al. IEEE TMI 2021] by randomly splitting MBs' whole tracks. In addition, only regions with vascular signals were used for FRC analysis, as the FRC method might be biased for very under-sampled data [Hingot et al. IEEE TMI 2021].

Below figure shows the difference between FRC analysis with the two splitting method. Note that the revised resolution is larger, but still below half a wavelength.

Figure. Image resolution estimated by FRC between two splitting methods. a, MB pairings between two frames were randomly split into two groups. b, MB whole trajectories across frames were randomly split into two groups following Hingot et al. 2021.

We revised the FRC analysis with the Hingot 2021 splitting method and added all the FRC curves into the Supplementary Figures.

Fig. F9] FSC curves. a, Short-axis view of the ex vivo porcine heart. b, short-axis view of the Patient 1. c, short-axis view of the Patient 2. d, long-axis view of the ex vivo porcine heart. e, long-axis view of the Patient 1. f, short-axis view of the patient 3. g, short-axis view of the patient 4. The arrows pointed the cross between the FRC curve and the half bit threshold curve, and the numbers indicate the corresponding image resolution measured by the method.

Below content has been revised in the main text.

"Image resolution was also estimated by a Fourier ring correlation (FRC)-based method through randomly splitting MBs' whole tracks across frames in two subsets, following a previous study [temp]. Based on the FRC estimation with ½ bit information threshold, the imaging resolution for ex vivo short- and long-axis views were 132 and 173 μm, the resolution for Patient 1 in short- and long-axis views were 149 and 202 μm, the resolution for short-axis view of Patient 2 was 154 μm, the resolution for short-axis view of Patient 3 was 240 μm, and the resolution for short-axis view of Patient 4 was 153 μm. FSC curves are provided in Fig.F9 in the Supplementary Figures. The above estimated resolutions are below half wavelength of the transmission pulses (320 μm for 2.4 MHz and 452 μm for 1.7 MHz). The variations of the estimated resolution in different images using FRC are likely due to the saturation level of the vascular images is low and hence causing uncertainties in FRC estimation [temp] "

[temp] Hingot, V., Chavignon, A., Heiles, B. & Couture, O. Measuring Image Resolution in Ultrasound Localization Microscopy. IEEE Trans. Med. Imaging 40, 3812–3819 (2021).

We added flow speed profiles across arteriole sections in Fig. 5. We cannot do the quantitative analysis between flows in diastole and systole like figure 3e-f in Demene et al. Nature BME 2021, because the vessels in the porcine heart were supplied by pump and only frames in diastole phases of patients were used.

Fig. 5 | Ex vivo porcine heart and in vivo patient short-axis view of flow speed, direction, flow speed profile. a, d, the flow speed maps, corresponding to the SR density map in Fig. 2 and 3. b, c, SR flow direction map of the zoomed-in regions inside the white boxed in a and b. e, f the flow speed profiles at the positions indicated by the red lines in b and c, which were generated by speeds on segments averaged across 0.5 mm width along the vessels.

We add below content in the Supplementary Figures, the saturation curve, to state the limited saturation in our study.

Fig. F13 | Saturation curve of the short-axis data of Patient 1 versus the acquisition time. Maps with pixel size of half or a quarter wavelength (λ) were filled by localisations. Saturation was calculated by the ratio of filled area to all the area in the myocardium. The figure demonstrates that the SR images are not fully saturated, as even the curve plotted with pixel size of half wavelength, which is double the size we used for smoothing, does not become flat with increasing acquisition time.

2. A major limitation of ultrasound localization microscopy is that the absence of a vessel can either be due to a perfusion defect, which has diagnostic value, or to an incomplete mapping of vascular networks by microbubbles, which is a technical limitation. Given the relatively short acquisition times reported here (10 cardiac cycles max), you cannot generate complete maps of the coronary vasculature. How will you dissociate impaired myocardial microcirculation from incomplete mapping

of vessels due to short acquisition times? Your images are still far from revealing the complete coronary vasculature. This should be discussed.

Response: We fully acknowledge that the ULM generally does not fully sample the whole vascular network, and it is even more so in our case when the sample time is very limited due to the breath holding constraints. Below sentence has been revised to clarify the limitation and potential improvement.

“The acquisition time of the cardiac data is orders of magnitude less than those typically used e.g., in brain ULM (Errico 2015, Heiles et al. Nature BME 2022 and Demene et al. Nature BME 2021) where no breath holding and cardiac gating were required. This short acquisition time results in limited saturation of vasculature in the reconstructed cardiac SRUS images, as demonstrated in Fig. F13 in the Supplementary Figures, and makes it challenging to distinguish impaired microcirculation from incomplete mapping of vessels. 3D imaging with longer acquisition and 3D motion correction could help address this issue, which is discussed later.”

3. I am puzzled by flow profiles reported in Figure 5a. I would have expected higher flow velocities in the epicardial coronary segments than in the penetrating arterioles. Can you provide the upper and lower flow velocity limits that can be detected with your current pipeline, and discuss the impact of vessel orientation on your results?

Response: We agree with the reviewer that higher velocity is expected in some larger epicardial vessels. We provided an additional video, i.e., Supplementary Video 5, demonstrating the localisation and tracking on the data corresponding to Figure 5a. Upon examining the video, flow velocities in the epicardial coronary segments, zoom-in region in Figure 5a, seems not significantly higher than the penetrating arterioles. This could well be due to vessel orientations in 3D while the imaging is only tracking velocities in 2D, hence highlighting the importance of moving to 3D. The upper flow velocity limit of our processing is, 150 mm/s, due to the searching window size used for tracking MBs. The lower limit is determined by the moving average subtraction as a high-pass filter. Our velocity map shows that some flow as slow as 5mm/s can be detected.

We add below content into Result section to state factors affecting detected flow.

“There might be bias in measured flows. The 2D ultrasound can only detect the velocity projected to the imaging plane where elevational component of velocity is lost. The maximum detectable velocity in the processing was restricted by the searching window size used in tracking MBs, which was set as 150 mm/s.”

4. Data processing steps to generate your supplementary video are not described. What period of the cardiac cycle are you showing precisely? Is the movie displaying consecutive diastoles or are you looping over a single averaged diastole? You have to ensure reproducibility of your results.

Response: We apologise for having not made this clear. Below video caption has been added into the supplementary material.

“Image/video sequence in Supplementary video 1 in the sequence of appearance: 1) Acquisition position demonstration. 2) B-mode frames of two consecutive cardiac cycles. 3) CEUS frames of the two consecutive full cardiac cycles obtained with moving-average subtraction and DAS beamforming. 4) CEUS frames in four consecutive diastoles obtained with angle MoCo and CV beamforming. 5) SR density maps. 6) Comparison with CTCA. 7) Microcirculation demonstration.

Minor comments:

1. Remove super-resolution from the title, ultrasound localization microscopy is enough and more accurate.

Response: Removed.

2. Remove SRUS from the text and stick to ULM.

Response: We have removed SRUS from the rest of the text, except for in the beginning where both ULM and SRUS are defined, as both have been used in previous literatures.

3. Line 47: this sentence is wrong. ULM is a class of super-resolution methods among others. Please correct.

Response: We revised the sentence as below.

“Super-resolution ultrasound (SRUS) through localising and tracking bubbles, also known as ultrasound localisation microscopy (ULM), can image deep microvasculature with resolution beyond the diffraction limit by accurately localising microbubbles (MBs) from the contrast-enhanced ultrasound (CEUS) images.”

4. Line 96: describe data processing performed on the RF data in supplementary figure 1 or another supplementary figure, this is interesting.

Response: We have done as suggested and please find the details in our response to Comment C4.

5. Line 102: can you quantify and report myocardial motion in your ex vivo and in vivo datasets? You could use tissue Doppler imaging for example.

Response: The motion range for the ex vivo and in vivo hearts were estimated by the image registration, and shown in the response to Comment C1 in section 1.

6. Figure 2: why is the apical vasculature not detected (Figure 2e)? Where was the cross section of the heart taken, at the basal, mid or apical level (Figure 2c)?

Response: We believe there was flow impairments in the apical region. Please see more detailed answer to the first question in the response to Comment C2. The cross-section in Fig. 2c was taken in the mid-level. The myocardium became thicker as the heart was submerged into the Tyrode's solution for hours and gradually swelling.

7. Lines 202-203: the claim about improved management of cardiac patients is not supported by your data.

Response: This has now been removed.

8. The benefit of nonlinear Doppler is not immediately clear to me, please expand. You could refer to efforts to design more specific ultrasound pulse sequences.

Response: Nonlinear tissue residual artefact in CEUS image can be due to tissue motion – if tissue moves between the contrast pulse sequence then the linear cancellation won't work. Nonlinear Doppler has the advantage of being able to take the motion into account and separate moving tissues from moving bubbles. While it is not expected to be helpful for this study as we deliberately chose diastole only when tissue motion is small, but systole phases of the in vivo data could be used in the future where Nonlinear Doppler could be helpful. Below sentence has been revised in the main text for clarification.

“The sensitivity, specificity and SNR of the MB images could be improved by e.g. using nonlinear Doppler [Temp] when tissue motion is significant, as it is capable of separating tissue and bubble signals even when they are moving.”

[Temp] Tremblay-Darveau, C., Williams, R., Milot, L., Bruce, M. & Burns, P. N. Visualizing the Tumor Microvasculature With a Nonlinear Plane-Wave Doppler Imaging Scheme Based on Amplitude Modulation. *IEEE Trans. Med. Imaging* 35, 699–709 (2016).

9. Line 233, cite Demeulenaere et al. *JACC Imaging* 2022.

Response: Cited.

10. Line 265. Beside the centre frequency, can you indicate the transmit frequency you used for your AM sequence?

Response: Below sentence has been added into the Method Section.

“A transmit frequency of 2.4 MHz was used for *ex vivo* experiment and Patient 1, and 1.7 MHz was used for Patient 2, 3 and 4.”

11. Report the transmitted pressure as a function of depth in a supplementary figure.

Response: Below measured pressure curves has been added into the Supplementary Figures.

Fig. F11 | Pressure at different depths along the central line of the probe. The 2.4 MHz was transmitted at 7 V, and 1.7 MHz was transmitted at 8.2 V when acquiring data. Pressure was measured underwater by a needle hydrophone (0.2 mm, Precision Acoustic, UK) for both the used transmitted frequencies respectively. The measured pressure was derated by 0.3 dB/MHz/cm to get the curves in the figure, considering sound attenuation in tissues.

12. Report estimated PSFs as a function of depth in a supplementary figure.

Response: Below sentence has been added into Method sections.

“PSFs estimated from *ex vivo* data were presented in Fig. F6 of the Supplementary Figures.”

Below estimated PSFs has been added in the supplementary figure.

Fig. F6 | PSF estimated from the *ex vivo* data (axis unit: mm). Note that PSFs were estimated from MB images that were segmented after removing noise by thresholding, resulting in seemingly smaller PSFs in regions with low SNR include the side and bottom areas.. The patches were cropped to the estimated PSF tightly to save computation in normalised cross-correlation.

Reviewer #3

The paper by Yan et al. reports of super-resolution ultrasound techniques in order to visualize microvascular cardiac disease in patients with a simple echocardiogram. The data presented are very compelling and demonstrate the potential of this technique to be used clinically for the early detection of heart disease.

We thank the reviewer for the encouraging comments.

One shortcoming that could be addressed by the authors is to pinpoint the methodology that made it this imaging possible. This is because all the techniques presented such as MoCo, SVD etc. have previously been implemented by the authors and others so it is not clear whether a specific breakthrough needed to be accomplished. Was it the fuzzy initialization? The AM sequence? If so, why? Some important methodological details are lacking which would aid further in the assessment of the performance of the technique. The transthoracic implementation is exciting but not clear why it was possible in the first place. Along those lines, results before and after different implementations would be helpful. Finally, some syntax and grammatical errors appear occasionally throughout which should be easily fixed.

Response: we believe that it is the combination of the various key elements of the pipeline, instead of a specific element, that made this possible. The key elements of the pipeline include: a carefully designed clinical acquisition protocol (including optimised bubble concentration), a high frame rate data acquisition (305 fps), sensitive and contrast specific pulsing sequence (three pulse AM in this case), high performance beamforming (Coherence based beamforming in our case), multi-level motion correction algorithms, and SRUS post-processing including localisation and tracking. Below figure has been added into the Supplementary Material to demonstrate that the used techniques, motion correction and CV beamformer, both contributed to improvement of image quality.

Fig. F2 | Power Doppler images of CEUS sequence acquired from the short-axis view of Patient 1 and reconstructed with different methods. **a**, image of the CEUS sequence used for SR processing. **b**, Image of the CEUS sequence obtained by replacing our CV beamforming with the conventional DAS beamforming and coherent compounding. By comparing **a** and **b**, CV beamforming can significantly reduce side lobes and noise. **c**, Image of CEUS sequence by removing Tissue Motion Correction (MoCo) from **b**. The vessels were significantly blurred without Tissue MoCo.

Below figure was provided into the Supplementary Figure to illustrate the contribution of Doppler-based motion correction on MBs to the contrast.

Fig. F1 | Evidence of improvement by using Doppler-based MoCo for compounding, noted as Angle Moco for convenience. **a**, compounded images reconstructed by DAS with and without Angle MoCo were normalised, log compressed with dynamic range of 40dB, and then overlaid by Matlab ‘imfuse’ function, using red for the image without Angle MoCo, green for image with Angle Moco, and yellow for areas of similar intensity between the two images. Green colour cover more area, at which there seems to be single bubbles. **b**, magnified image of the region in red box in **a**. Images were plotted in polar coordinates, linear scale and same dynamic range ([0, 4]), after envelope detection. Compounded image obtained by averaging angle image with Angle MoCo presents stronger intensity than its counterpart. **c**, real part of the signals along the yellow line in **b**. Signal shifts among steering angles can be corrected and thus gives a higher average.

We added below figure into the supplementary material to demonstrate contribution of the proposed tracking framework by comparison with the SimpleTracker used in two previous ULM studies. (in Heiles et al. Nature BME 2022 and Demene et al. Nature BME 2021)

Fig. F4| Velocity magnitude map obtained by our proposed method and the “SimpleTracker(tinevez/simpletracker (MathWorks)”. “SimpleTracker” as set with no gap filling, the same searching window size, 150 mm/s, and the same filtering, only accepting MBs with persistence more than 3 frames. **a**, map obtained by our tracking framework. **b**, magnified image of the region in the dashed line of **a**. **c**, map obtained by the “SimpleTracker”. **d**, magnified image of the region in the dashed line of **c**. **b** provides visually clearer vasculature than **d**. “SimpleTracker” pairs MB by finding total minimum of cost that is defined with distance between MBs. It prefers to pair closer MBs but also accepts pairs with larger distance only if two MBs are within the searching window and satisfy the topology constraint. The trajectories, pointed by white arrows in **d** are likely to be errors, as they are between vessel branches and with speed much higher than surroundings. These trajectories were rejected by the proposed method, as motion model worked as an additional constraint.

1. Line 38 – ‘CMD is also associated...’ Please provide references for the statements made here.
 Response: Below reference is cited in this line.

[temp] Kunadian, Vijay, et al. "An EAPCI expert consensus document on ischaemia with non-obstructive coronary arteries in collaboration with European Society of Cardiology Working Group on Coronary Pathophysiology & Microcirculation Endorsed by Coronary Vasomotor Disorders International Study Group." *European heart journal* 41.37 (2020): 3504-3520.

2. Line 74- Please be more specific regarding the type of ‘impaired left ventricular function’ that the patients in this study were diagnosed with.

Response: We have added more specific diagnosis for the four patients. The sentence in the has been revised as below.

“The feasibility of the pipeline was demonstrated on two *ex vivo* Langendorff porcine hearts, and on four patients (the first one with impaired left ventricular function from the hypertrophic cardiomyopathy, the second one with a dilated left ventricle secondary to high ventricular ectopy burden, the third one with idiopathic ventricular fibrillation in a structurally normal heart, and the fourth one with hypertrophic cardiomyopathy with asymmetric septal hypertrophy).”

More details of patients can be found in the method section.

“Patient 1 was a 60-year-old patient (male, BMI 24 kg/m² – 167 cm, 67 kg) with a hypertrophic cardiomyopathy and secondary prophylactic left-sided dual chamber implantable cardioverter defibrillator (ICD). Patient 2 was a 31-year-old patient (male, BMI 23.3 kg/m² – 191 cm, 83.0 kg) with

dilated left ventricular cavity (139.57ml/m², but preserved left Ventricular Ejection Fraction (LVEF): 63%), no regional wall motion abnormalities, without evidence of myocardial scar in the cardiac MRI. The left ventricular dilatation was likely secondary to a high burden of ventricular ectopy of >40%. Patient 3 was a 31-year-old patient (male, BMI 25.3 kg/m² – 175 cm, 77.6 kg) with background of idiopathic ventricular fibrillation with structurally normal heart in cardiac MRI and secondary prevention transvenous dual chamber ICD insertion who presented with appropriate ICD shocks due to recurrent ventricular ectopic triggered ventricular fibrillation. Patient 4 was a 34-year-old patient (male, BMI 28.9 kg/m² - 178.5cm, 72.8kg) with hypertrophic cardiomyopathy with asymmetric septal hypertrophy (max. wall thickness 21mm) without outflow tract obstruction, LVEF 75%, and primary prevention subcutaneous ICD. Cardiac MRI reported patchy Late Gadolinium Enhancement (LGE) throughout the hypertrophied septum as well as large dense perfusion defects throughout the septum in during perfusion study with adenosine.”

3. Line 87 – How was the AM sequence used here to separate the bubble signal? Please specify all parameters and rationale used and whether this was a key aspect of the technique.

Response: The details of AM and rationale were added to the manuscript as below. We believe a good contrast pulsing sequence is a key aspect of the technique as having good quality contrast specific data is a pre-requisite for generating super-localisation and tracking images. Below sentences have been added in the method section to state the used sequence and the rationales.

“While the Verasonics system took time to change its output voltage and there was potential nonlinearity in the output voltage of the system and in the response of piezoelectric elements, the ultrasound fields with full or half amplitudes in the AM sequence were not generated by exciting the probe with different voltages. Instead, a three pulse AM sequence with half, full, and half amplitude. The ultrasound fields with half amplitude were generated by exciting half number of elements and interleaved groups, each of which consisting of two elements in the probe, as shown in Fig. 1, to achieve improved linear cancellation and reduce the impact of cross-talk between neighbouring elements[temp]. The pulse central frequency and transmitted pressure were between 50 and 90 kPa at the depth between 30 and 50 mm, as demonstrated in Fig. F11 in in the Supplementary Figures. The transmitted pressure was chosen so that we obtain good SNR without significantly destroying the bubbles. The pulse length is kept as short as possible to maintain spatial resolution. Two different transmit frequencies 1.7 and 2 MHz were used. The lower frequency has higher bubble detection sensitivity, but 2MHz has slightly better resolution.”

[temp] Lai, Ting-Yu, and Michalakis A. Averkiou. "Contrast-Enhanced Ultrasound with Optimized Aperture Patterns and Bubble Segmentation Based on Echo Phase." *Ultrasound in Medicine & Biology* 49.1 (2023): 186-202.

Fig. F11 | Pressure at different depths along the central line of the probe. The 2.4 MHz was transmitted at 7 V, and 1.7 MHz was transmitted at 8.2 V when acquiring data. Pressure was measured underwater by a needle hydrophone (0.2 mm, Precision Acoustic, UK) for both the used transmitted frequencies respectively. The measured pressure was derated by 0.3 dB/MHz/cm to get the curves in the figure, considering sound attenuation in tissues.

4. Line 89- please specify the concentration, size distribution and composition of the bubbles used here. Was the dosage used here the clinical dose?

Response: The concentration of Sonovue bubbles is around 2.0×10^8 per millilitre. Below figure presents the size distribution provided by Bracco. Bubble's gas core is Sulphur Hexafluoride and the shell is Phospholipid molecule. More details can be found in https://www.ema.europa.eu/en/documents/product-information/sonovue-epar-product-information_en.pdf. The dosage used is within the clinical range.

Fig. F10 | Diameter distribution of Sonovue microbubbles. PDF: probability density function

We revised the sentences as below.

“Sonovue (Bracco, Milan, Italy) MBs, whose concentration is around 2×10^8 per millilitre, gas core is Sulphur Hexafluoride, shell is Phospholipid molecule and size distribution can be found in Fig. F10 of the Supplementary Figures, were used as ultrasound contrast agents in this study.”

“For the *in vivo* human heart imaging, 2 ml of MBs, which was within the recommended clinical dose range, were manually injected in a slow bolus (around 6 seconds).”

5. Lines 96 and 106: Please provide all methodological details for the AM sequence and the SVD filtering, respectively.

Response: Methodological details for the AM sequence can be found in the response to Comment C4. Below citation was added for the SVD filtering in the line. More details can be found in the Method Section 3 and 4.

Demené, Charlie, et al. "Spatiotemporal clutter filtering of ultrafast ultrasound data highly increases Doppler and fUltrasound sensitivity." *IEEE transactions on medical imaging* 34.11 (2015): 2271-2285.

Below sentence in the Method section was revised to describe the SVD filtering.

"B-mode ultrasound images in each cardiac cycle were processed by MATLAB function, 'svd', and reconstructed with the 5% largest singular values to reduce the effect of moving MB signals on motion estimation."

6. Lines 116- how was the Kalman motion model used here? Please provide all important methodological details.

Response: The sentence has been revised below to summarise how Kalman motion model was used. We also provided a five-page methodological description about the Kalman motion model in the Supplementary Method.

"MBs were paired with our previously proposed feature-motion-model framework[temp1], where the ratio of normalised image intensity difference between candidate MB pairs to probability obtained from linear Kalman motion model with MB locations was set as cost function and MBs were paired by finding total minimum with graph-based assignment [temp2]."

"MBs were detected to appear no more than 3 frames were regarded as low confidence tracks and removed to improve tracking precision."

[temp1] Yan, J., Zhang, T., Broughton-Venner, J., Huang, P. & Tang, M.-X. Super-Resolution Ultrasound Through Sparsity-Based Deconvolution and Multi-Feature Tracking. *IEEE Trans. Med. Imaging* 41, 1938–1947 (2022).

[temp2] Sbalzarini, I. F. & Koumoutsakos, P. Feature point tracking and trajectory analysis for video imaging in cell biology. *J. Struct. Biol.* **151**, 182–195 (2005)

7. Fig. 2e. There seems to be an angle dependence of the technique in the Langerdorff model, the lateral walls are very dark. Please indicate whether this is a limitation of the technique or it's in vitro implementation.

Response: We do believe that the absence of vascular flow in the apical area is likely due to impaired flow, as well as some sensitivity drop off near the image boundary. Please find the detailed response to Comment C2.

8. Line 130-'myocardium surround the left ventricle...' Please correct for syntax/grammar. Response: Thanks for pointing out the grammar. The sentence has been corrected as below.

"myocardium surrounding the left ventricle..."

9. Fig. 3d. There seem to be dark regions in the image, similar to the in vitro case. Is this due to the

divergent beamforming sequence? Please provide an explanation on the potential angle dependence.
Response: Yes the dark regions may also be caused by angle dependence of probe, due to the element directivity, as shown below.

Fig. F12 | Probe sensitivity. The map was obtained by summing all the element sensitivity to each pixel. The element sensitivity can be described as $\text{sinc}\left(\frac{L}{\lambda}\sin(\theta)\right) \times \cos(\theta)$, where L is the element width; λ is the wavelength; θ is the angle between the axial direction and the line connecting the element centre and pixel.

Below sentence has been added into the result section.

“Myocardial vessels close to the top-right margin of heart in Fig. 3d might not be detected due to reduced probe sensitivity close to left and right boundaries that affects both transmission and reception and is shown in the Fig. F12 of the Supplementary Figure.”

10. Line 158-Please indicate the epicardium and endocardium on the figure.

Response: We pointed out each layer in the figure Fig3c and Fig4a as below.

Fig. 3 | *In vivo* CTCA scans and SRUS imaging for patient one.

Fig. 4 | *In vivo* CTCA scan and SRUS images for patient two.

11. Figure 3- Please show images before and after MoCo to better evaluate the effect of the technique. What was the range of motion estimated *in vivo*? How did it compare to *in vitro*?

Response: Fig. F2b and c, presented in the response to your first comment, show the difference between power Doppler images with and without motion correction. It can be seen that clearer vasculature can be obtained after MoCo. Motion range in the in vivo diastole phase and the in vitro whole cycle were approximate 1.7 and 1.4 mm, respectively. Please refer to response to Comment C1 in Section 1 to find more details.

12. Line 180- How were these vessels validated? Please indicate how ICM could be evaluated using this technique.

Response: By comparing the CTCA and SRUS images in Fig. 3 and Fig. 4, SRUS can reconstruct large vessels in CTCA and more small vessels that are not presented in CTCA. It is challenging to validate these small vessels in vivo as no existing non-invasive method has the sensitivity and resolution required. However, as microbubbles are pure vascular agents and do not extravasate, we have high degree of certainty that each microbubble location indicates the present of a blood vessel containing this bubble.

The term ischemic cardiomyopathy refers to structural and functional alterations of the myocardial microvasculature secondary to poor blood supply, leading to ischemia and myocardial infarction. Our technology may be applied to ICM patients for a number of possible scenarios:

Firstly, structural microvasculature disease endotype may represent architectural changes, such as fibrosis. The ability to visualise the coronary vessels in high spatial resolution would allow for the direct presenting of structural changes of the vessel, and could provide non-invasive assessment of functional via flow and velocity measurements at the site of the narrowing and in the downstream vasculature.

Secondly, at sites remote to the fibrosis, compensatory hypertrophic changes of the healthy myocardium may occur with increased resting blood flow, which could be quantified via the flow velocity and vessel diameter measured from by SRUS/ULM technique as an additional objective measurement to judge degree of severity of the disease.

Added content can be found in the response to your next comment.

13. Line 183: What is the significance of quantifying coronary flow velocity? Could this aid in identification of ischemia and/or infarction or treatment thereof?

Response: At sites of coronary artery stenosis, flow velocity will increase. Also, a direct visualisation of the myocardial vasculature to quantify vessel dilatation and flow volume at rest and during stress may provide a diagnostic tool to directly assess the functional relevance of coronary artery disease.

The functionally relevant epicardial coronary stenosis induces compensatory vasodilatation in the downstream microvascular segments to maintain myocardial flow at rest. As such coronary flow reserve may be exhausted already at rest and not allow for appropriate augmentation during stress, and results in ischemia. This technique would allow to assess coronary flow reserve to identify such coronary microvasculature disease.

An animal 3D study showed the ability of ULM in assessing myocardial infarction resulting from coronary artery disease [temp1].

[Temp 1] Demeulenaere, O. et al. Coronary Flow Assessment Using 3-Dimensional Ultrafast Ultrasound Localization Microscopy. JACC Cardiovasc. Imaging 15, 1193–1208 (2022).

We cannot claim the significance in diagnosis and treatment of those cardiac patients yet based on the current data, due to the limited number of cases. Although more evidence and research are required to derive tailored treatment strategies from this information, it is reasonable to assume that additional diagnostic findings will impact clinical decision making and patient management.

Below content into Introduction Section has been revised to state a traditional way for diagnosing Cardiac Microvascular Disease (CMD).

“Comparing to large artery disease, our understanding of CMD is still very limited. Traditionally CMD is diagnosed by documentation of a diminished coronary flow reserve and impaired ability of the microvascular to augment blood flow in response to stress, which is indirectly evaluated from microvascular flow in arteries [Temp 1]. Initially CMD was thought to be a combination of structural and functional changes at the level of the microvasculature, recent studies have led to a more differentiated understanding and sub-classification to structural CMD and functional CMD subtypes [Temp 2,3].”

[Temp 1] Sinha, A., Rahman, H. & Perera, D. Coronary microvascular disease: current concepts of pathophysiology, diagnosis and management. *Cardiovasc. Endocrinol. Metab.* 10, 22–30 (2020).

[Temp 2] Rahman, H. et al. Coronary Microvascular Dysfunction Is Associated With Myocardial Ischemia and Abnormal Coronary Perfusion During Exercise. *Circulation* 140, 1805–1816 (2019).

[Temp 3] Rahman, H. et al. Physiological Stratification of Patients With Angina Due to Coronary Microvascular Dysfunction. *J. Am. Coll. Cardiol.* 75, 2538–2549 (2020).

We add below sentence into Introduction Section to state the potential application of SRUS/ULM.

“The concept of using ULM for imaging coronary vasculature in myocardium has been recently demonstrated in small animals in 2D and 3D [temp 1,2]. The 3D study showed the ability in assessing myocardial infarction resulting from coronary artery disease. Microvascular disease is an increasingly recognised important differential diagnosis to ischemia caused by epicardial coronary artery disease. Despite its clinical importance, in human assessment has so far largely been restricted to indirect measurements of myocardial blood flow and coronary flow reserve. Direct visualisation of myocardial vasculature at a microcirculation-level and hemodynamic quantification via ULM to investigate and characterise structural and functional microcirculation alterations would fill an important gap in contemporary cardiological diagnostics.”

[Temp 1] Demeulenaere, O. et al. Coronary Flow Assessment Using 3-Dimensional Ultrafast Ultrasound Localization Microscopy. *JACC Cardiovasc. Imaging* 15, 1193–1208 (2022).

[Temp 2] Cormier, P., Porée, J., Bourquin, C. & Provost, J. Dynamic Myocardial Ultrasound Localization Angiography. *IEEE Trans. Med. Imaging* 40, 3379–3388 (2021).

14. Figure 4. Similar dark regions in the lateral wall in (b). Please explain.

Response: Please find the response to Comment C3.

15. Line 200: Please quantify the improvement over the diffraction limit.

Response: The description of the resolution was also updated in the manuscript:

"The cross-section analysis was also performed on two visually separated vessels in the magnified regions (Fig. 3e and 3f). As shown in Fig. 3h and 3i, these two pairs of vessels are 302.1 μm and 267.8 μm apart, compared to half the wavelength of 320 μm ."

"Image resolution was also estimated by a Fourier ring correlation (FRC)-based method through randomly splitting MBs' whole tracks across frames in two subsets, following a previous study [temp]. Based on the FRC estimation with $\frac{1}{2}$ bit information threshold, the imaging resolution for ex vivo short- and long-axis views were 132 and 173 μm , the resolution for Patient 1 in short- and long-axis views were 149 and 202 μm , the resolution for short-axis view of Patient 2 was 154 μm , the resolution for short-axis view of Patient 3 was 240 μm , and the resolution for short-axis view of Patient 4 was 153 μm . FSC curves are provided in Fig.F9 in the Supplementary Figures. The above estimated resolutions are below half wavelength of the transmission pulses (320 μm for 2.4 MHz and 452 μm for 1.7 MHz). The variations of the estimated resolution in different images using FRC are likely due to the saturation level of the vascular images is low and hence causing uncertainties in FRC estimation [temp]"

[temp] Hingot, V., Chavignon, A., Heiles, B. & Couture, O. Measuring Image Resolution in Ultrasound Localization Microscopy. IEEE Trans. Med. Imaging 40, 3812–3819 (2021).

Fig. F9 | FSC curves. **a**, Short-axis view of the ex vivo porcine heart. **b**, short-axis view of the Patient 1. **c**, short-axis view of the Patient 2. **d**, long-axis view of the ex vivo porcine heart. **e**, long-axis view of the Patient 1. **f**, short-axis view of the patient 3. **g**, short-axis view of the patient 4. The arrows pointed the cross between the FRC curve and the half bit threshold curve, and the numbers indicate the corresponding image resolution measured by the method.

16. Line 223: Please indicate whether the clinical findings were in agreement with the clinical diagnosis and why.

Response: Currently our understanding of the myocardial vascular geometry and flow dynamics in normal and pathological conditions is limited, as there is until now no effective in vivo imaging tools. One thing that is known clinically is that for patients with myocardial hypertrophy, coronary vasodilator reserve is characteristically reduced due to a variety of structural and functional changes including vascular dilatation to increase resting coronary flow to maintain tissue oxygenation. For the

HCM patient we recruited in this study, the super resolution imaging visualised many dilated vessels in keeping with his clinical diagnosis.

We added below sentences into the Result Section.

“Dilated myocardial vessels can be directly visualised from the reconstructed vasculature of Patient 1 diagnosed with hypertrophic cardiomyopathy. This type of cardiomyopathy is associated with a reduced coronary vasodilator reserve and increased resting coronary flow to meet the higher baseline oxygen demand due to the elevated myocardial mass and higher filling pressure [temp].

[temp] Anderson, H. V. et al. Coronary Artery Flow Velocity Is Related To Lumen Area and Regional Left Ventricular Mass. *Circulation* 102, 48–54 (2000)

Rebuttal 2

RE: Decision on Article nBME-23-0750A

Dear Editor and Reviewers,

The authors are grateful for the advice and comments. We have listed our responses and revisions in this document, which we hope would address the concerns and questions raised by the reviewers.

Texts in the manuscript are given between “Double quotation marks” in this letter, and revised texts are highlighted by yellow colour.

Reviewer #1:

The author has answered most questions and revised the manuscript. However, ULM technique has been proposed over a decade with both technique and application advances, including animal brain imaging (Errico et al., Nature 2015), 4D mapping (Rabut et al., Nature Methods, 2019), functional ULM (Renaudin et al., Nature Methods, 2022) and patient study (Demene et al., Nature BME, 2021). It seems that this study only shifts existing ULM technique to another application – myocardial vasculature. In addition, the clinical benefit of using ULM is not clear since no extra information was provided in this study. Overall, the advances in both clinical impact and ULM methodology are questionable.

Response: We respectfully disagree. The reviewer listed some significant publications in ULM. However, we would like to point out that the Errico 2015 Nature paper is not the first in vivo ULM study (See Siepmann et al., IEEE IUS, 2011 & Christensen-Jeffries et al., IEEE TMI 2015 which was available online in 2014, all demonstrated ULM in mice). Furthermore, the Demene 2021 nBME paper is not the first in patient work (Opacic et al. Nature Communications 2018 demonstrated ULM in breast cancer patients). However they still deserve being published in their respective journals as they have demonstrated the feasibility of ULM in vivo and in human in an important organ which is the brain. In this study, we have demonstrated ULM in arguably one most challenging organ in human, the heart, due to its large motion, requiring a suite of technological developments as detailed in our manuscript including a multi-level motion correction framework (correction between steering angles, 2 stage correction between frames within a cardiac cycle, and between different cardiac cycles), combined with new tracking initialisation method and coherence based beamforming. As the second reviewer has pointed out, we are reporting the highest resolution ultrasound images of the coronary vasculature in patients to date.

As suggested by the last-round comments, we do not claim any direct clinical benefit due to limited data. However, this study paves the way for assessing the myocardial vasculature and further clinical impact can be made with clinical trials, given the importance of these intra-myocardial vessels. Up to 25% of patients with coronary artery disease symptoms have no stenosis in large coronary vessels and have micro-vessel abnormalities. ULM as demonstrated in this study can provide more understanding of the cardiac microvascular disease and has potential to impact the decision making and monitoring of treatment of this important disease.

1. One key advantage of ULM technique is to provide much higher resolution. However, the reconstructed ULM image quality in this study is poor. For instance, the quantified resolution is ranging from 150 to 250 μm which is very close to the half-wavelength of the transmission pulse (320 μm for 2.4 MHz transducer). Therefore, the advantages of using ULM are questionable and should be carefully addressed. Small resolution improvements could be realized by other approaches instead of time-consuming ULM processing.

Response: We acknowledge that the improvement of resolution compared with half a wavelength is not as much as some existing studies on other organs. However, as commented by the second reviewer, it is the improvement over existing B-mode resolution that matters in the end. The diffraction limit is the theoretically best resolution achievable by conventional imaging and the actual resolution of existing echocardiography in patients is much larger than this theoretical limit. The initial image lateral resolution as measured by the FWHM of PSF is around 1700(1950), 3971(4520) and 5850(7310) μm at depth of 30, 80 and 120 mm for frequency 2.4 (1.7MHz) respectively, and therefore our ULM image resolution presents improvements ranging from 8 to 30 folds. Below sentence has been added in the results to address the significance of using ULM technology.

“The lateral resolution of the contrast echocardiograph, as measured by the FWHM of the PSF, was 1700 (1950) and 5850 (7310) μm at depth of 30 and 120 mm for transmitted frequency 2.4 (1.7) MHz respectively. The ULM image resolutions as estimated by FRC are just below half a wavelength of the transmission pulses (320 μm for 2.4 MHz and 452 μm for 1.7 MHz) and present at least 8-fold improvement to the contrast echocardiography.”

2. The flow speed of the myocardial vasculature has a huge variation during the cardiac cycle. Does the presented ULM technique have the capability to cover such a large range? If so, please provide the calibration results and its comparison with the ground truth (i.e., use optical image as standard). In addition, the effective frame rate is only 305 Hz with limited acquisition time, how does the author ensure the accuracy of your ULM reconstruction? The images presented in this study are averaged over time, right? It will lose temporal information which is a drawback of ULM but important for heart observation.

Response: We agree with the reviewer that the range of flow speed during a full cardiac cycle is large. We are not claiming that the technique covers the full range of cardiac vascular flow. In our study we only used data in the cardiac diastolic phase where the heart muscle relaxes. The images in this study are accumulated and speed averaged over this cardiac phase.

A frame rate of 305 Hz is empirically chosen to maintain a balance between frame rate and the quality of the compounded contrast images. The limited total acquisition time is dictated by the length the patients can hold their breath. We fully acknowledge these limitations and that the final ULM images only present a small sample of the vast microvasculature within the organ.

We have revised below statement in the discussion to clarify the limitation.

“While ULM has revealed both macro- and micro-vessels in the myocardium in the four patients, these are only a small sample of the large vascular network in the myocardium, and there are regions with no apparent vessels identified. In this study the total data acquisition time is limited to $\sim 10\text{s}$, which is dictated by the length that patients can hold their breath. Only data from the diastolic phase of the cardiac cycle with least myocardium motion was used for generating ULM images. Therefore, only vessels and flows perfused in this phase were reconstructed. There might be bias in measured flows. A frame rate of 305Hz and searching window size, set as 150 mm/s, used in tracking MBs would also mean that the fastest flow in large vessels could be missed. The 2D ultrasound can only detect the velocity projected to the imaging plane where elevational component of velocity is lost. Temporal information was not present in the ULM images due to the temporal accumulation process in the reconstruction.”

“ Future work to develop ultrafast 3D cardiac ultrasound can facilitate motion correction and further improve the myocardial ULM. Correction of 3D cardiac motion is feasible if 3D + time imaging can be achieved using e.g. a matrix array transducer. With whole cardiac cycle data available, temporally

resolved flow speed can be reconstructed to show pulsatility [temp1], and presented via a microbubble density cineloop [temp2].

[temp1] Bourquin, C., Porée, J., Lesage, F. & Provost, J. In Vivo Pulsatility Measurement of Cerebral Microcirculation in Rodents Using Dynamic Ultrasound Localization Microscopy. *IEEE Trans. Med. Imaging* 41, 782–792 (2022).

[temp2] Cormier, P., Porée, J., Bourquin, C. & Provost, J. Dynamic Myocardial Ultrasound Localization Angiography. *IEEE Trans. Med. Imaging* 40, 3379–3388 (2021)

While only using data from the diastolic phase would reduce the flow speed range and make tracking task easier, it is not possible to validate this in human as there is currently no other modality capable of tracking such microvascular flow. To offer some crude idea as how the tracking works on our data, we have generated an additional Supplementary Video 6, where it shows the tracking of individual bubbles in a human myocardium.

“Demonstrations of MB tracking can be found in Supplementary Video 5 and 6.”

Supplementary Video 6: Demonstration of localisation and tracking at the top-left region on short-axis view CEUS frames of Patient 1. Right: red crosses denote localised positions of MBs. Left: Flow speed map with temporally accumulated MB trajectories after persistence filtering. Dynamic range of CEUS images: 100 dB.

3. MB concentration is an important parameter in ULM reconstruction. How does the author control the accuracy of MB localization and tracking under a high MB concentration as well as a high flow range? Please specific it in detail. The author should also explain why there was limited detectable vessels in the images which is contradiction to the heart structure (plenty of vessels). In addition, results are presented for the diastolic phase, why not for a full cardiac cycle?

Response: We have revised below sentence to clarify the purpose of our injection strategy.

“We injected MBs through either infusion or a slow bolus, instead of a fast bolus, to avoid too high MB concentrations in myocardium, as localising and tracking MBs can be more challenging at higher concentration. For the ex vivo porcine heart imaging, MBs were infused by a syringe pump (Harvard Apparatus, Holliston, MA, USA) at an infusion rate of 5 ml/min. For the in vivo human heart imaging, 2 ml of MBs, which was within the recommended clinical dose range, were manually injected in a slow bolus (around 6 seconds). ”

With above injection and image settings, the apparent MB concentration were not as high, as demonstrated by the Supplementary Video 5 and 6. Further to our response to your last comment, we have revised below sentence in the discussion to clarify the limitation of our localisation method.

“Normalised cross-correlation, smoothing CEUS images with PSFs, was chosen as the localisation method to deal with limited SNR in CEUS images but reduced the probability of separating closely located bubbles. This might cause nonnegligible errors in the reconstruction of microvessels with a diameter of a few hundred microns or those that are separated by a few hundred microns. Therefore, we used infusion and slow bolus to inject MBs rather than the fast bolus in this study, to reduce MB concentration and close MBs.”

Given the limited data acquisition time, the ULM images only present a small sample of the full microvasculature in the organ and we have acknowledged this as shown in our response to the last question and below. “This short acquisition time results in limited saturation of vasculature in the

reconstructed cardiac ULM images, as demonstrated in Fig. F13 in the Supplementary Figures, and makes it challenging to distinguish impaired microcirculation from incomplete mapping of vessels.”

The reason we only did ULM on diastolic phase is to avoid large cardiac motion, as diastolic phase has the least motion in the full cardiac cycle. We revised below sentences to state the reason clearer in Results and Methods sections.

“For the in vivo human dataset specifically, an image-intensity-based gating algorithm was firstly implemented to select and index frames in the diastolic phase, which were with least motions among the whole cardiac cycle, so as to avoid significant out-of-plane motion.”

“As large out-of-plane motion correction was challenging for 2D ultrasound due to loss of signals, we focused on the diastole, where twisting motion is the least among all the phases in a cardiac cycle.”

As stated in the response to your last question, 3D imaging could resolve the problem by capturing the motions that were out of imaging plane of 2D ultrasound.

4. Apart from other organs – liver, kidney, brain, motion artifacts of heart could significantly downgrade the ULM quality even hold the breathing. Although authors implemented motion registration algorithms to compensate its morphological changes, the rapid heart volume alternation will largely affect the MB distribution, resulting in uncertain trajectory, out-of-plane movement and so forth. How does the author address this issue?

Response: As we only use the diastolic phase of the data, we do not have the issue of rapid heart volume alternation. We acknowledge that this would be an important issue if dynamic ULM is to be achieved in human hearts.

5. If this study is designed for a new application – cardiac disease, please highlight the significance of quantifying small vessels. According to the presented ULM image, the resolution improvement is not enough (from 320 μm in theoretical to $\sim 250 \mu\text{m}$ in measurement). In addition, the advantages over conventional imaging such as CTCA, functional MRI is not explained in this study. Moreover, many methodological details have been proposed by the author’s group and others. The reviewer is not clear about the technique advance. Is there any novel approach applied to make this cardiac vasculature feasible? Please highlights this in detail.

Response: We have below sentence in the discussion to highlight the significance

“The high resolution and sensitivity to myocardial vascular flow and its quantification potentially leads to better understanding of microcirculations in the myocardium.”

We cannot claim more significance than the above in the manuscript as suggested by the last-round review due to the limited data. Instead, we have stated some potential significance in the introduction.

“The concept of using ULM for imaging coronary vasculature in myocardium has been recently demonstrated in small animals in 2D and 3D. The 3D study showed the ability in assessing myocardial infarction resulting from coronary artery disease. Microvascular disease is an increasingly recognised important differential diagnosis to ischemia caused by epicardial coronary artery disease. Despite its clinical importance, in human assessment has so far largely been restricted to indirect measurements of myocardial blood flow and coronary flow reserve. Direct visualisation of myocardial vasculature at a microcirculation-level and hemodynamic quantification via ULM to investigate and characterise

structural and functional microcirculation alterations would fill an important gap in contemporary cardiological diagnostics.”

Our image resolution improvement over the standard contrast echo is significant (>8 folds). As reviewer two has pointed out, this improvement is what matters. We have also generated the highest resolution images of in human myocardial vasculature, as pointed out by reviewer two.

As demonstrated in Fig. 3&4 and Supplementary Fig. F8, CTCA presents much less small vessels for its lower sensitivity than the ULM and MRI is additionally limited in the resolution. The comparison between CTCA and ULM has been shown in the Result section. We added below sentence in the Result section to emphasize the advantages of ULM over CTCA and MRI.

“In short, visual comparison demonstrates ULM images present overall higher resolution and sensitivity to small myocardial vessels.”

Besides being the first demonstration of cardiac ULM in human, we also have technical advances including a comprehensive multi-level motion correction pipeline different from our previous one to deal with image sequence, as well as contrast image enhancement methods using the combination of AM sequence, Doppler-based motion correction for angle compounding, and CV beamforming. We have also proposed the use of Fuzzy Initialisation and Parameter Estimation in our tracking strategy, which are described in our five-page Supplementary Method.

Reviewer #2:

1. Please include FRC curves in one of the main figures of the manuscript rather than in supplementary, this is proper metric to report resolution. Even though you "only" reach $\lambda/2$, you are still reporting the highest resolution ultrasound images of the coronary vasculature in patients to date. Pay attention in the text, it is FRC and not FSC, please correct.

Response: Thanks for your positive comments. We have added the FRC curves in the main text and corrected the typo.

Fig. 5 | FRC curves. a, evaluation on the short-axis view of five datasets. b, evaluation on the long-axis view of two datasets. The arrows point at the crosses between the FRC curve and the half bit threshold curve, and the numbers indicate the corresponding image resolution measured by the method.

2. Can you add to the text a comparison of your FRC resolution and B-mode resolution? This is what matters in the end and it will compare more favorably that the wavelength.

Response: Thanks for your suggestions. We added below sentence in the result and the abstract.

“Due to the usage of diverging wave, the implemented ultrafast contrast echocardiography has increasing size of PSF along the imaging depth, whose lateral FWHM was measured to be around 1700 (1950) and 5850 (7310) μm at depth of 30 and 120 mm for transmitted frequency 2.4 (1.7) MHz respectively. The above resolutions estimated by FRC are below half wavelength of the transmission pulses (320 μm for 2.4 MHz and 452 μm for 1.7 MHz) and present at least 8-fold improvement to the contrast echocardiograph.”

3. Given the reported FRC resolution, I would not claim in the abstract that you detect vessels beyond the resolution limit but at the resolution limit ($\lambda/2$). Please correct.

Response: Corrected.

4. Figure 5, please perform a bin analysis of the velocity profiles as in Errico et al. Nature, 2015 figure 2E.

Response: We added the bin analysis as below figure. Considering the limited number of MBs due to the limited acquisition time, we separated the profile into three bins to get enough samples for statistical analysis. As the speed distribution was tested not to be normal, unpaired Mann-Whitney U-test was used to test the significance among bins.

Fig. 5 | Ex vivo porcine heart and in vivo patient short-axis view of flow speed, direction, flow speed profile. a, d, the flow speed maps, corresponding to the SR density map in Fig. 2 and 3. b, c, SR flow direction map of the zoomed-in regions inside the white boxed in a and b. e, f, the flow speed profiles at the positions indicated by the red lines in b and c, which were generated by speeds on segments averaged across 0.5 mm width along the vessels. g, h, Flow speed distribution of MBs passing through red lines in b and c. Bin width: 200 μm ; black line: median; white square: mean, box: 25th and 75th percentile; whiskers: min and max data points excluding outliers. Unpaired Mann-Whitney U-test was used. ** $P < 0.01$. e, f, g and h demonstrate that large vessels have higher flow speed at the centre than the side.

5. I still have a question regarding the data processing steps to generate your supplementary video. What do you mean by "7) Microcirculation demonstration"? Please describe these processing steps in the text.

Response: We revised the step as "Microcirculation Visualisation" and added below sentence to the video caption for clarification.

"Microcirculation visualisation was done by moving localised microbubbles along their trajectories. Each scatter in the visualisation represents one localised microbubble."

6. Remove "SRUS/ULM" from the text and simply use ULM throughout the manuscript. Ultrasound localization microscopy (ULM) is a subclass of vascular super resolution ultrasound (SRUS) methods. Your sentence line 48 is still not accurate. Please rephrase as: "Ultrasound localization microscopy (ULM), a class of super-resolution ultrasound (SRUS) methods, can image deep microvasculature [...]"

Response: Revised.

Reviewer #3:

The authors did an excellent job responding to the comments with new figures, new videos and new supplementary data. Some of the details of the method regarding improvements still remain unclear especially regarding the details of the AM approach and the MoCo method *in vivo* so it is highly recommended that the authors increase the rigor by providing parameters that they believed increased the image quality reported.

Response: Thanks for your recognition. We have added one new table in the Supplementary Material to summarise the parameters and components used in the data acquisition and processing, and also added below sentence in the caption of Fig. 1.

“Parameters and components used for acquiring and processing *in vivo* data are summarized in Supplementary Table S1.”

Table S1 | Parameters used for *in vivo* data acquisition and processing. More details can be found in the Method Section and Supplementary Method.

	Parameter/ Component	Value/ Brief Description	
Acquisition with GEM5ScD probe	Transmitted centre frequency	2.4 MHz or 1.7 MHz	
	Diverging wave focus	-21.6 mm	
	Steering angle sequence	-15°, -3°, 9°, 15°, 3°, -9°	
	Driving voltage	7 V or 8.2 V, measured pressures are shown in Fig. F10	
	AM sequence (Half-Full-Half aperture)	Interleaved groups with two elements, i.e., driving elements as (11001100...) or (00110011...) for half amplitude.	
	Frame rate	305 Hz	
	Duration	10 second	
CEUS reconstruction with MB MoCo (Diagram shown in Fig. F5)	Polar grid	67.8 μm and 0.5°	
	Doppler window size	1 mm and 3.5°	
Multi-level Motion Correction	Image difference metric	Sum of square pixel intensity difference	
	Tissue signal extraction	Keep 5% largest singular values of SVD for reconstruction	
	Intra-cycle tissue MoCo (two-stage registration: affine + B-spline-based non-rigid)	B-mode image dynamic range	50 dB after log compression
		Cartesian grid	Depth: 67.8 μm. Lateral: 135.0 μm
		B-spline grid	Depth: 2.17 mm. Lateral: 4.32 mm
		Solver	Affine: Levenberg–Marquardt algorithm Non-rigid:

		Steepest decent algorithm
	Maximum iteration steps	500 times
	Stopping criteria	Image difference between two adjacent iterations changed less than 0.001% for 20 times.
Inter-cycle tissue MoCo (Rigid registration)	Averaged CEUS dynamic range	100 dB after log compression
	The other parameters were same with the affine registration used for the intra-cycle motion correction	
Localisation	Cartesian grid	13.5 μm in both directions
	Threshold on normalised cross-correlation coefficient map	0.5
Tracking	Searching window size	150 mm/s
	Kalman parameters	Estimated from the data. Details can be found in the supplementary method.
	MB persistence filtering	>3 frames